# Challenging Common Assumptions in the Unsupervised Learning of Disentangled Representations

Francesco Locatello[2,3], Stefan Bauer[3], Mario Lucic[1], Gunnar Rätsch[2], Sylvain Gelly[1], Bernhard Schölkopf[3], and Olivier Bachem[1]

[1]Google AI, Brain team
[2]ETH Zurich, Dept. for Computer Science
[3]Max-Planck Institute for Intelligent Systems

## Abstract

The key idea behind the *unsupervised* learning of *disentangled* representations is that real-world data is generated by a few explanatory factors of variation which can be recovered by unsupervised learning algorithms. In this paper, we provide a sober look on recent progress in the field and challenge some common assumptions. We train more than $12\,000$ models covering most prominent methods and evaluation metrics in a reproducible large-scale experimental study on seven different data sets. We observe that while the different methods successfully enforce properties "encouraged" by the corresponding losses, well-disentangled models seemingly cannot be identified without supervision. Furthermore, increased disentanglement does not seem to lead to a decreased sample complexity of learning for downstream tasks. Our results suggest that future work on disentanglement learning should be explicit about the role of inductive biases and (implicit) supervision, investigate concrete benefits of enforcing disentanglement of the learned representations, and consider a reproducible experimental setup covering several data sets.

## 1 Introduction

In representation learning it is often assumed that real-world observations $\mathbf{x}$ (e.g., images or videos) are generated by a two-step generative process. First, a multivariate latent random variable $\mathbf{z}$ is sampled from a distribution $P(\mathbf{z})$. Intuitively, $\mathbf{z}$ corresponds to semantically meaningful factors of variation of the observations (e.g., content + position of objects in an image). Then, in a second step, the observation $\mathbf{x}$ is sampled from the conditional distribution $P(\mathbf{x}|\mathbf{z})$. The key idea behind this model is that the high-dimensional data $\mathbf{x}$ can be explained by the substantially lower dimensional and semantically meaningful latent variable $\mathbf{z}$ which is mapped to the higher-dimensional space of observations $\mathbf{x}$. Informally, the goal of representation learning is to find useful transformations $r(\mathbf{x})$ of $\mathbf{x}$ that "make it easier to extract useful information when building classifiers or other predictors" [4]. A recent line of work has argued that representations that are *disentangled* are an important step towards a better representation learning [4, 46, 36, 3, 51, 33, 58]. They should contain all the information present in $\mathbf{x}$ in a compact and interpretable structure [4, 31, 8] while being independent from the task at hand [17, 37]. They should be useful for (semi-)supervised learning of downstream tasks, transfer and few shot learning [4, 52, 46]. They should enable to integrate out nuisance factors [32], to perform interventions, and to answer counterfactual questions [44, 53, 46].

While there is no single formalized notion of disentanglement (yet) which is widely accepted, the key intuition is that a disentangled representation should separate the distinct, informative *factors*

*of variations* in the data [4]. A change in a single underlying factor of variation $z_i$ should lead to a change in a single factor in the learned representation $r(\mathbf{x})$. This assumption can be extended to groups of factors as, for instance, in Bouchacourt et al. [5] or Suter et al. [56]. Based on this idea, a variety of disentanglement evaluation protocols have been proposed leveraging the statistical relations between the learned representation and the ground-truth factor of variations. Disentanglement is then measured as a particular structural property of these relations [19, 29, 15, 32, 7, 49].

State-of-the-art approaches for unsupervised disentanglement learning are largely based on *Variational Autoencoders (VAEs)* [30]: One assumes a specific prior $P(\mathbf{z})$ on the latent space and then uses a deep neural network to parameterize the conditional probability $P(\mathbf{x}|\mathbf{z})$. Similarly, the distribution $P(\mathbf{z}|\mathbf{x})$ is approximated using a variational distribution $Q(\mathbf{z}|\mathbf{x})$, again parametrized using a deep neural network. The model is then trained by minimizing a suitable approximation to the negative log-likelihood. The representation for $r(\mathbf{x})$ is usually taken to be the mean of the approximate posterior distribution $Q(\mathbf{z}|\mathbf{x})$. Several variations of VAEs were proposed with the motivation that they lead to better disentanglement [19, 6, 29, 7, 32, 50]. The common theme behind all these approaches is that they try to enforce a factorized aggregated posterior $\int_{\mathbf{x}} Q(\mathbf{z}|\mathbf{x})P(\mathbf{x})d\mathbf{x}$, which should encourage disentanglement. In Appendix A we theoretically prove that (perhaps unsurprisingly) the unsupervised learning of disentangled representations is fundamentally impossible without inductive biases both on the considered learning approaches and the data sets.

**Our contributions.** In light of this result, we challenge commonly held assumptions in this field. Our contributions can be summarized as follows:

- We investigate current approaches and their inductive biases in a reproducible large-scale experimental study[1] with a sound experimental protocol for unsupervised disentanglement learning. We implement six recent unsupervised disentanglement learning methods as well as six disentanglement measures from scratch and train more than 12 000 models on seven data sets.

- We release `disentanglement_lib`[2], a new library to train and evaluate disentangled representations. As reproducing our results requires substantial computational effort, we also release more than 10 000 trained models which can be used as baselines for future research.

- We analyze our experimental results and challenge common beliefs in unsupervised disentanglement learning: (i) While all considered methods prove effective at ensuring that the individual dimensions of the aggregated posterior (which is sampled) are not correlated, we observe that the dimensions of the representation (which is taken to be the mean) are correlated. (ii) We do not find any evidence that the considered models can be used to reliably learn disentangled representations in an *unsupervised* manner as random seeds and hyperparameters seem to matter more than the model choice. Furthermore, good trained models seemingly cannot be identified without access to ground-truth labels even if we are allowed to transfer good hyperparameter values across data sets. (iii) For the considered models and data sets, we cannot validate the assumption that disentanglement is useful for downstream tasks, for example through a decreased sample complexity of learning.

- Based on these empirical evidence, we suggest three critical areas of further research: (i) The role of inductive biases and implicit and explicit supervision should be made explicit: unsupervised model selection persists as a key question. (ii) The concrete practical benefits of enforcing a specific notion of disentanglement of the learned representations should be demonstrated. (iii) Experiments should be conducted in a reproducible experimental setup on data sets of varying degrees of difficulty.

## 2 Other related works

In a similar spirit to disentanglement, (non-)linear independent component analysis [12, 2, 27, 24] studies the problem of recovering independent components of a signal. The underlying assumption is that there is a generative model for the signal composed of the combination of statistically independent non-Gaussian components. While the identifiability result for linear ICA [12] proved to be a milestone for the classical theory of factor analysis, similar results are in general not obtainable for the nonlinear case and the underlying sources generating the data cannot be identified [25]. The lack of almost any

---

[1]Reproducing these experiments requires approximately 2.52 GPU years (NVIDIA P100).
[2]https://github.com/google-research/disentanglement_lib

identifiability result in non-linear ICA has been a main bottleneck for the utility of the approach [26] and partially motivated alternative machine learning approaches [14, 51, 11]. Given that unsupervised algorithms did not initially perform well on realistic settings most of the other works have considered some more or less explicit form of supervision [47, 63, 61, 31, 9, 39, 42, 56]. [22, 10] assume some knowledge of the effect of the factors of variations even though they are not observed. One can also exploit known relations between factors in different samples [28, 18, 60, 16, 13, 23, 62] or explicit inductive biases [38]. This is not a limiting assumption especially in sequential data, i.e., for videos. We focus our study on the setting where factors of variations are not observable at all, i.e. we only observe samples from $P(\mathbf{x})$.

## 3   Experimental design

**Considered methods.** All the considered methods augment the VAE loss with some regularizer. The $\beta$-VAE [19], introduces a hyperparameter in front of the KL regularizer of vanilla VAEs to constrain the capacity of the VAE bottleneck. The AnnealedVAE [6] progressively increase the bottleneck capacity so that the encoder can focus on learning one factor of variation at the time (the one that most contribute to a small reconstruction error). The FactorVAE [29] and the $\beta$-TCVAE [7] penalize the total correlation [59] with adversarial training [43, 55] or with a tractable but biased Monte-Carlo estimator respectively. The DIP-VAE-I and the DIP-VAE-II [32] both penalize the mismatch between the aggregated posterior and a factorized prior. Implementation details and further discussion on the methods can be found in Appendix C and H.

**Considered metrics.** The *BetaVAE* metric [19] measures disentanglement as the accuracy of a linear classifier that predicts the index of a fixed factor of variation. Kim & Mnih [29] address several issues with this metric in their *FactorVAE* metric by using a majority vote classifier on a different feature vector which accounts for a corner case in the BetaVAE metric. The *Mutual Information Gap (MIG)* [7] measures for each factor of variation the normalized gap in mutual information between the highest and second highest coordinate in $r(\mathbf{x})$. Instead, the *Modularity* [49] measures if each dimension of $r(\mathbf{x})$ depends on at most a factor of variation using their mutual information. The Disentanglement metric of Ridgeway & Mozer [49] (which we call *DCI Disentanglement* for clarity) computes the entropy of the distribution obtained by normalizing the importance of each dimension of the learned representation for predicting the value of a factor of variation. The *SAP score* [32] is the average difference of the prediction error of the two most predictive latent dimensions for each factor. Implementation details and further descriptions can be found in Appendix D.

**Data sets.** We consider four data sets in which $\mathbf{x}$ is obtained as a deterministic function of $\mathbf{z}$: *dSprites* [19], *Cars3d* [48], *SmallNORB* [35], *Shapes3D* [29]. We also introduce three data sets where the observations $\mathbf{x}$ are stochastic given the factor of variations $\mathbf{z}$: *Color-dSprites*, *Noisy-dSprites* and *Scream-dSprites*. In *Color-dSprites*, the shapes are colored with a random color. In *Noisy-dSprites*, we consider white-colored shapes on a noisy background. Finally, in *Scream-dSprites* the background is replaced with a random patch in a random color shade extracted from the famous *The Scream* painting [40]. The dSprites shape is embedded into the image by inverting the color of its pixels. Further details on the preprocessing of the data can be found in Appendix I.

We fix our experimental setup in advance and we run all the considered methods on each data set for 50 different random seeds and evaluate them on the considered metrics. The full details on the experimental setup are provided in the Appendix H. Our experimental conditions and guiding principles, the limitations of this study, and the differences with previous implementations are extensively discussed in Appendix E, F, and G respectively.

## 4   Key experimental results

In this section, we highlight our key findings with plots specifically picked to be representative of our main results. In Appendix J, we provide the full experimental results with a complete set of plots for different methods, data sets and disentanglement metrics.

## 4.1 Can current methods enforce a uncorrelated aggregated posterior and representation?

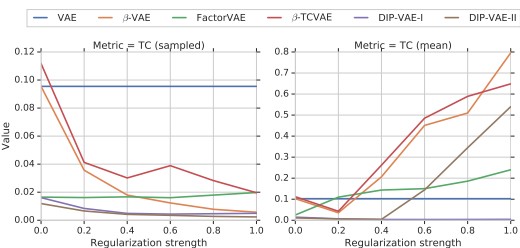

Figure 1: Total correlation based on a fitted Gaussian of the sampled (left) and the mean representation (right) plotted against regularization strength for Color-dSprites and approaches (except AnnealedVAE). The total correlation of the sampled representation decreases while the total correlation of the mean representation increases as the regularization strength is increased.

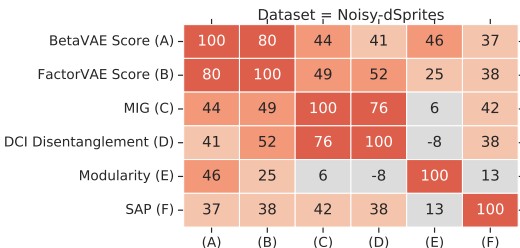

Figure 2: Rank correlation of different metrics on Noisy-dSprites. Overall, we observe that all metrics except Modularity seem mildly correlated with the pairs BetaVAE and FactorVAE, and MIG and DCI Disentanglement strongly correlated with each other.

While many of the considered methods aim to enforce a factorizing and thus uncorrelated aggregated posterior (e.g., regularizing the total correlation of the sampled representation), they use the mean vector of the Gaussian encoder as the representation and not a sample from the Gaussian encoder. This may seem like a minor, irrelevant modification; however, it is not clear whether a factorizing aggregated posterior also ensures that the dimensions of the mean representation are uncorrelated. To test the impact of this, we compute the total correlation of both the mean and the sampled representation based on fitting Gaussian distributions for each data set, model and hyperparameter value (see Appendix D and J.2 for details).

Figure 1 (left) shows the total correlation based on a fitted Gaussian of the *sampled* representation plotted against the regularization strength for each method except AnnealedVAE on Color-dSprites. We observe that the total correlation of the sampled representation generally decreases with the regularization strength. One the other hand, Figure 1 (right) shows the total correlation of the *mean* representation plotted against the regularization strength. It is evident that the total correlation of the mean representation generally increases with the regularization strength. The only exception is DIP-VAE-I for which we observe that the total correlation of the mean representation is consistently low. This is not surprising as the DIP-VAE-I objective directly optimizes the covariance matrix of the mean representation to be diagonal which implies that the corresponding total correlation (as we measure it) is low. These findings are confirmed by our detailed experimental results in Appendix J.2 (in particular Figures 8-9) which considers all different data sets. Furthermore, we observe largely the same pattern if we consider the average mutual information between different dimension of the representation instead of the total correlation (see Figures 26-27 in Appendix K).

**Implications.** Overall, these results lead us to conclude with minor exceptions that the considered methods are effective at enforcing an aggregated posterior whose individual dimensions are not correlated but that this does not seem to imply that the dimensions of the mean representation (usually used for representation) are uncorrelated.

## 4.2 How much do the disentanglement metrics agree?

As there exists no single, common definition of disentanglement, an interesting question is to see how much different proposed metrics agree. Figure 2 shows the Spearman rank correlation between different disentanglement metrics on Noisy-dSprites whereas Figure 12 in Appendix J.3 shows the correlation for all the different data sets. We observe that all metrics except Modularity seem to be correlated strongly on the data sets dSprites, Color-dSprites and Scream-dSprites and mildly on the other data sets. There appear to be two pairs among these metrics that capture particularly similar notions: the BetaVAE and the FactorVAE score as well as the MIG and DCI Disentanglement.

**Implication.** All disentanglement metrics except Modularity appear to be correlated. However, the level of correlation changes between different data sets.

## 4.3 How important are different models and hyperparameters for disentanglement?

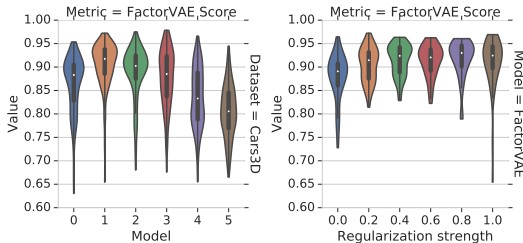

Figure 3: (left) FactorVAE score for each method on Cars3D. Models are abbreviated (0=$\beta$-VAE, 1=FactorVAE, 2=$\beta$-TCVAE, 3=DIP-VAE-I, 4=DIP-VAE-II, 5=AnnealedVAE). The scores are heavily overlapping. (right) Distribution of FactorVAE scores for FactorVAE model for different regularization strengths on Cars3D.

The primary motivation behind the considered methods is that they should lead to improved disentament. This raises the question how disentanglement is affected by the model choice, the hyperparameter selection and randomness (in the form of different random seeds). To investigate this, we compute all the considered disentanglement metrics for each of our trained models.

In Figure 3 (left), we show the range of attainable FactorVAE scores for each method on Cars3D. We observe that these ranges are heavily overlapping for different models leading us to (qualitatively) conclude that the choice of hyperparameters and the random seed seems to be substantially more important than the choice of objective function. These results are confirmed by the full experimental results on all the data sets presented in Figure 13 of Appendix J.4: While certain models seem to attain better maximum scores on specific data sets and disentanglement metrics, we do not observe any consistent pattern that one model is consistently better than the other. At this point, we note that in our study we have fixed the range of hyperparameters *a priori* to six different values for each model and did not explore additional hyperparameters based on the results (as that would bias our study). However, this also means that specific models may have performed better than in Figure 13 (left) if we had chosen a different set of hyperparameters.

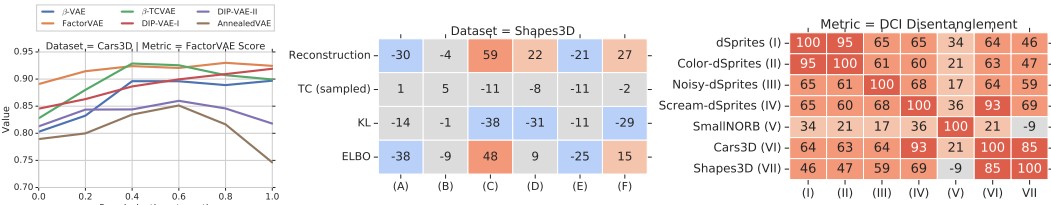

Figure 4: (left) FactorVAE score vs hyperparameters for each score on Cars3d. There seems to be no model dominating all the others and for each model there does not seem to be a consistent strategy in choosing the regularization strength. (center) Unsupervised scores vs disentanglement metrics on Shapes3D. Metrics are abbreviated ((A)=BetaVAE Score, (B)=FactorVAE Score, (C)=MIG , (D)=DCI Disentanglement, (E)=Modularity, (F)=SAP). The unsupervised scores we consider do not seem to be useful for model selection. (right) Rank-correlation of DCI disentanglement metric across different data sets. Good hyperparameters only seem to transfer between dSprites and Color-dSprites but not in between the other data sets.

In Figure 3 (right), we further show the impact of randomness in the form of random seeds on the disentanglement scores. Each violin plot shows the distribution of the FactorVAE metric across all 50 trained FactorVAE models for each hyperparameter setting on Cars3D. We clearly see that randomness (in the form of different random seeds) has a substantial impact on the attained result and that a good run with a bad hyperparameter can beat a bad run with a good hyperparameter in many cases. Again, these findings are consistent with the complete set of plots provided in Figure 14 of Appendix J.4.

Finally, we perform a variance analysis by trying to predict the different disentanglement scores using ordinary least squares for each data set: If we allow the score to depend only on the objective function (treated as a categorical variable), we are only able to explain 37% of the variance of the scores on average (see Table 5 in Appendix J.4 for further details). Similarly, if the score depends on the Cartesian product of objective function and regularization strength (again categorical), we are able to explain 59% of the variance while the rest is due to the random seed.

**Implication.** The disentanglement scores of unsupervised models are heavily influenced by randomness (in the form of the random seed) and the choice of the hyperparameter (in the form of the regularization strength). The objective function appears to have less impact.

## 4.4 Are there reliable recipes for model selection?

In this section, we investigate how good hyperparameters can be chosen and how we can distinguish between good and bad training runs. In this paper, we advocate that that model selection *should not* depend on the considered disentanglement score for the following reasons: The point of unsupervised learning of disentangled representation is that there is no access to the labels as otherwise we could incorporate them and would have to compare to semi-supervised and fully supervised methods. All the disentanglement metrics considered in this paper require a substantial amount of ground-truth labels or the full generative model (for example for the BetaVAE and the FactorVAE metric). Hence, one may substantially bias the results of a study by tuning hyperparameters based on (supervised) disentanglement metrics. Furthermore, we argue that it is not sufficient to fix a set of hyperparameters *a priori* and then show that one of those hyperparameters and a specific random seed achieves a good disentanglement score as it amounts to showing the existence of a good model, but does not guide the practitioner in finding it. Finally, in many practical settings, we might not even have access to adequate labels as it may be hard to identify the true underlying factor of variations, in particular, if we consider data modalities that are less suitable to human interpretation than images.

In the remainder of this section, we hence investigate and assess different ways how hyperparameters and good model runs could be chosen. In this study, we focus on choosing the learning model and the regularization strength corresponding to that loss function. However, we note that in practice this problem is likely even harder as a practitioner might also want to tune other modeling choices such architecture or optimizer.

**General recipes for hyperparameter selection.** We first investigate whether we may find generally applicable "rules of thumb" for choosing the hyperparameters. For this, we plot in Figure 4 (left) the FactorVAE score against different regularization strengths for each model on the Cars3D data set whereas Figure 16 in Appendix J.5 shows the same plot for all data sets and disentanglement metrics. The values correspond to the median obtained values across 50 random seeds for each model, hyperparameter and data set. Overall, there seems to be no model consistently dominating all the others and for each model there does not seem to be a consistent strategy in choosing the regularization strength to maximize disentanglement scores. Furthermore, even if we could identify a good objective function and corresponding hyperparameter value, we still could not distinguish between a good and a bad training run.

**Model selection based on unsupervised scores.** Another approach could be to select hyperparameters based on unsupervised scores such as the reconstruction error, the KL divergence between the prior and the approximate posterior, the Evidence Lower BOund or the estimated total correlation of the sampled representation (mean representation gives similar results). This would have the advantage that we could select specific trained models and not just good hyperparameter settings whose median trained model would perform well. To test whether such an approach is fruitful, we compute the rank correlation between these unsupervised metrics and the disentanglement metrics and present it in Figure 4 (center) for Shapes3D and in Figure 16 of Appendix J.5 for all the different data sets. While we do observe some correlations, no clear pattern emerges which leads us to conclude that this approach is unlikely to be successful in practice.

Table 1: Probability of outperforming random model selection on a different random seed. A random disentanglement metric and data set is sampled and used for model selection. That model is then compared to a randomly selected model: (i) on the same metric and data set, (ii) on the same metric and a random different data set, (iii) on a random different metric and the same data set, and (iv) on a random different metric and a random different data set. The results are averaged across 10 000 random draws.

|  | Random data set | Same data set |
|---|---|---|
| Random metric | 54.9% | 62.6% |
| Same metric | 59.3% | 80.7% |

**Hyperparameter selection based on transfer.** The final strategy for hyperparameter selection that we consider is based on transferring good settings across data sets. The key idea is that good hyperparameter settings may be inferred on data sets where we have labels available (such as dSprites) and then applied to novel data sets.

Figure 4 (right) shows the rank correlations obtained between different data sets for the DCI disentanglement (whereas Figure 17 in Appendix J.5 shows it for all data sets). We find a strong and consistent correlation between dSprites and Color-dSprites. While these results suggest that some transfer of hyperparameters is possible, it does not allow us to distinguish between good and bad random seeds on the target data set.

To illustrate this, we compare such a transfer based approach to hyperparameter selection to random model selection as follows: First, we sample one of our 50 random seeds, a random disentanglement metric and a data set and use them to select the hyperparameter setting with the highest attained score. Then, we compare that selected hyperparameter setting to a randomly selected model on either the same or a random different data set, based on either the same or a random different metric and for a randomly sampled seed. Finally, we report the percentage of trials in which this transfer strategy outperforms or performs equally well as random model selection across 10 000 trials in Table 1. If we choose the same metric and the same data set (but a different random seed), we obtain a score of 80.7%. If we aim to transfer for the same metric across data sets, we achieve around 59.3%. Finally, if we transfer both across metrics and data sets, our performance drops to 54.9%.

**Implications.** Unsupervised model selection remains an unsolved problem. Transfer of good hyperparameters between metrics and data sets does not seem to work as there appears to be no unsupervised way to distinguish between good and bad random seeds on the target task.

## 4.5 Are these disentangled representations useful for downstream tasks in terms of the sample complexity of learning?

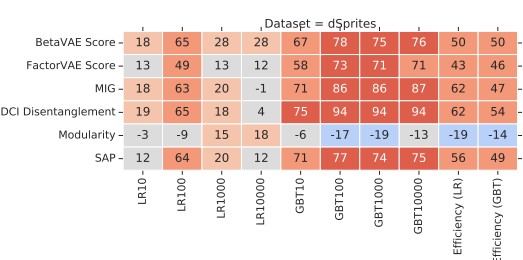

Figure 5: Rank correlations between disentanglement metrics and downstream performance (accu-

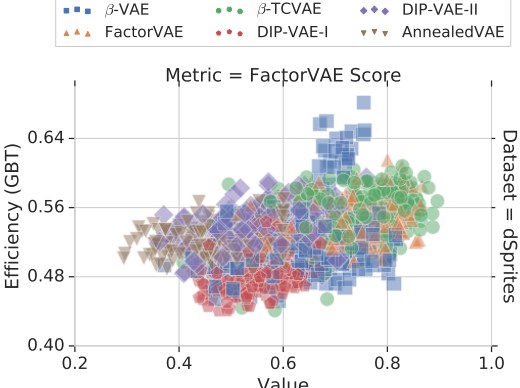

Figure 6: Statistical efficiency of the FactorVAE Score for learning a GBT downstream task on dSprites.

One of the key motivations behind disentangled representations is that they are assumed to be useful for later downstream tasks. In particular, it is argued that disentanglement should lead to a better sample complexity of learning [4, 52, 46]. In this section, we consider the simplest downstream classification task where the goal is to recover the true factors of variations from the learned representation using either multi-class logistic regression (LR) or gradient boosted trees (GBT).

Figure 5 shows the rank correlations between the disentanglement metrics and the downstream performance on dSprites. We observe that all metrics except Modularity seem to be correlated with increased downstream performance on the different variations of dSprites and to some degree on Shapes3D but not on the other data sets. However, it is not clear whether this is due to the fact that disentangled representations perform better or whether some of these scores actually also (partially) capture the informativeness of the evaluated representation. Furthermore, the full results in Figure 19 of Appendix J.6 indicate that the correlation is weaker or inexistent on other data sets (e.g. Cars3D).

To assess the sample complexity argument we compute for each trained model a statistical efficiency score which we define as the average accuracy based on 100 samples divided by the

average accuracy based on 10 000 samples. Figure 6 show the sample efficiency of learning (based on GBT) versus the FactorVAE Score on dSprites. We do not observe that higher disentanglement scores reliably lead to a higher sample efficiency. This finding which appears to be consistent with the results in Figures 20-23 of Appendix J.6.

**Implications.** While the empirical results in this section are negative, they should also be interpreted with care. After all, we have seen in previous sections that the models considered in this study fail to reliably produce disentangled representations. Hence, the results in this section might change if one were to consider a different set of models, for example semi-supervised or fully supervised one. Furthermore, there are many more potential notions of usefulness such as interpretability and fairness that we have not considered in our experimental evaluation. Nevertheless, we argue that the lack of concrete examples of useful disentangled representations necessitates that future work on disentanglement methods should make this point more explicit. While prior work [54, 34, 41, 20, 21] successfully applied disentanglement methods such as $\beta$-VAE on a variety of downstream tasks, it is not clear to us that these approaches and trained models performed well *because of disentanglement*.

## 5    Conclusions

In this work we first theoretically show that the unsupervised learning of disentangled representations is fundamentally impossible without inductive biases (Appendix A). We then performed a large-scale empirical study with six state-of-the-art disentanglement methods, six disentanglement metrics on seven data sets and conclude the following: (i) A factorizing aggregated posterior (which is sampled) does not seem to necessarily imply that the dimensions in the representation (which is taken to be the mean) are uncorrelated. (ii) Random seeds and hyperparameters seem to matter more than the model but tuning seem to require supervision. (iii) We did not observe that increased disentanglement implies a decreased sample complexity of learning downstream tasks. Based on these findings, we suggest three main directions for future research:

**Inductive biases and implicit and explicit supervision.** Our theoretical impossibility result in Appendix A highlights the need of inductive biases while our experimental results indicate that the role of supervision is crucial. As currently there does not seem to exist a reliable strategy to choose hyperparameters in the unsupervised learning of disentangled representations, we argue that future work should make the role of inductive biases and implicit and explicit supervision more explicit. We would encourage and motivate future work on disentangled representation learning that deviates from the static, purely unsupervised setting considered in this work. Promising settings (that have been explored to some degree) seem to be for example (i) disentanglement learning with interactions [57], (ii) when weak forms of supervision e.g. through grouping information are available [5], or (iii) when temporal structure is available for the learning problem. The last setting seems to be particularly interesting given recent identifiability results in non-linear ICA [24].

**Concrete practical benefits of disentangled representations.** In our experiments we investigated whether higher disentanglement scores lead to increased sample efficiency for downstream tasks and did not find evidence that this is the case. While these results only apply to the setting and downstream task used in our study, we are also not aware of other prior work that compellingly shows the usefulness of disentangled representations. Hence, we argue that future work should aim to show concrete benefits of disentangled representations. Interpretability and fairness as well as interactive settings seem to be particularly promising candidates to evaluate usefulness. One potential approach to include inductive biases, offer interpretability, and generalization is the concept of independent causal mechanisms and the framework of causal inference [44, 46].

**Experimental setup and diversity of data sets.** Our study also highlights the need for a sound, robust, and reproducible experimental setup on a diverse set of data sets in order to draw valid conclusions. We have observed that it is easy to draw spurious conclusions from experimental results if one only considers a subset of methods, metrics and data sets. Hence, we argue that it is crucial for future work to perform experiments on a wide variety of data sets to see whether conclusions and insights are generally applicable. This is particularly important in the setting of disentanglement learning as experiments are largely performed on toy-like data sets. We are hence interested in insights that generalize across multiple data sets rather than the absolute performance on specific data sets. For this reason, we released `disentanglement_lib`, the library we created to train and evaluate the different disentanglement methods end-to-end. We also released more than 10 000 trained models to provide a solid baseline for future methods and metrics.

## Acknowledgements

The authors thank Ilya Tolstikhin, Paul Rubenstein and Josip Djolonga for helpful discussions and comments. This research was partially supported by the Max Planck ETH Center for Learning Systems and by an ETH core grant (to Gunnar Rätsch). This work was partially done while Francesco Locatello was at Google AI.

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

# A  Impossibility result

The first question that we investigate is whether unsupervised disentanglement learning is even possible for arbitrary generative models. Theorem 1 essentially shows that without inductive biases both on models and data sets the task is fundamentally impossible. The proof is provided in Appendix B.

**Theorem 1.** *For $d > 1$, let $\mathbf{z} \sim P$ denote any distribution which admits a density $p(\mathbf{z}) = \prod_{i=1}^{d} p(\mathbf{z}_i)$. Then, there exists an infinite family of bijective functions $f : \mathrm{supp}(\mathbf{z}) \to \mathrm{supp}(\mathbf{z})$ such that $\frac{\partial f_i(\mathbf{u})}{\partial u_j} \neq 0$ almost everywhere for all $i$ and $j$ (i.e., $\mathbf{z}$ and $f(\mathbf{z})$ are completely entangled) and $P(\mathbf{z} \leq \mathbf{u}) = P(f(\mathbf{z}) \leq \mathbf{u})$ for all $\mathbf{u} \in \mathrm{supp}(\mathbf{z})$ (i.e., they have the same marginal distribution).*

Consider the commonly used "intuitive" notion of disentanglement which advocates that a change in a single ground-truth factor should lead to a single change in the representation. In that setting, Theorem 1 implies that unsupervised disentanglement learning is *impossible* for arbitrary generative models with a factorized prior[3] in the following sense: Assume we have $p(\mathbf{z})$ and some $P(\mathbf{x}|\mathbf{z})$ defining a generative model. Consider any unsupervised disentanglement method and assume that it finds a representation $r(\mathbf{x})$ that is perfectly disentangled with respect to $\mathbf{z}$ in the generative model. Then, Theorem 1 implies that there is an equivalent generative model with the latent variable $\hat{\mathbf{z}} = f(\mathbf{z})$ where $\hat{\mathbf{z}}$ is completely *entangled* with respect to $\mathbf{z}$ and thus also $r(\mathbf{x})$: as all the entries in the Jacobian of $f$ are non-zero, a change in a single dimension of $\mathbf{z}$ implies that all dimensions of $\hat{\mathbf{z}}$ change. Furthermore, since $f$ is deterministic and $p(\mathbf{z}) = p(\hat{\mathbf{z}})$ almost everywhere, both generative models have the same marginal distribution of the observations $\mathbf{x}$ by construction, i.e., $P(\mathbf{x}) = \int p(\mathbf{x}|\mathbf{z})p(\mathbf{z})d\mathbf{z} = \int p(\mathbf{x}|\hat{\mathbf{z}})p(\hat{\mathbf{z}})d\hat{\mathbf{z}}$. Since the (unsupervised) disentanglement method only has access to observations $\mathbf{x}$, it hence cannot distinguish between the two equivalent generative models and thus has to be entangled to at least one of them.

This may not be surprising to readers familiar with the causality and ICA literature as it is consistent with the following argument: After observing $\mathbf{x}$, we can construct infinitely many generative models which have the same marginal distribution of $\mathbf{x}$. Any one of these models could be the true causal generative model for the data, and the right model cannot be identified given only the distribution of $\mathbf{x}$ [46]. Similar results have been obtained in the context of non-linear ICA [25]. The main novelty of Theorem 1 is that it allows the explicit construction of latent spaces $\mathbf{z}$ and $\hat{\mathbf{z}}$ that are completely *entangled* with each other in the sense of [4]. We note that while this result is very intuitive for multivariate Gaussians it also holds for distributions which are not invariant to rotation, for example multivariate uniform distributions.

While Theorem 1 shows that unsupervised disentanglement learning is fundamentally impossible for arbitrary generative models, this does not necessarily mean it is an impossible endeavour in practice. After all, real world generative models may have a certain structure that could be exploited through suitably chosen inductive biases. However, Theorem 1 clearly shows that inductive biases are required both for the models (so that we find a specific set of solutions) and for the data sets (such that these solutions match the true generative model). We hence argue that the role of inductive biases should be made explicit and investigated further as done in our experimental study.

# B  Proof of Theorem 1

*Proof.* To show the claim, we explicitly construct a family of functions $f$ using a sequence of bijective functions. Let $d > 1$ be the dimensionality of the latent variable $\mathbf{z}$ and consider the function $g : \mathrm{supp}(\mathbf{z}) \to [0,1]^d$ defined by

$$g_i(\boldsymbol{v}) = P(\mathbf{z}_i \leq v_i) \quad \forall i = 1, 2, \ldots, d.$$

Since $P$ admits a density $p(\mathbf{z}) = \prod_i p(\mathbf{z}_i)$, the function $g$ is bijective and, for almost every $\boldsymbol{v} \in \mathrm{supp}(\mathbf{z})$, it holds that $\frac{\partial g_i(\boldsymbol{v})}{\partial v_i} \neq 0$ for all $i$ and $\frac{\partial g_i(\boldsymbol{v})}{\partial v_j} = 0$ for all $i \neq j$. Furthermore, it is easy to see that, by construction, $g(\mathbf{z})$ is a independent $d$-dimensional uniform distribution. Similarly, consider the function $h : (0,1]^d \to \mathbb{R}^d$ defined by

$$h_i(\boldsymbol{v}) = \psi^{-1}(v_i) \quad \forall i = 1, 2, \ldots, d,$$

---

[3]Theorem 1 only applies to factorized priors; however, we expect that a similar result can be extended to non-factorizing priors.

where $\psi(\cdot)$ denotes the cumulative density function of a standard normal distribution. Again, by definition, $h$ is bijective with $\frac{\partial h_i(\boldsymbol{v})}{\partial v_i} \neq 0$ for all $i$ and $\frac{\partial h_i(\boldsymbol{v})}{\partial v_j} = 0$ for all $i \neq j$. Furthermore, the random variable $h(g(\mathbf{z}))$ is a $d$-dimensional standard normal distribution.

Let $\boldsymbol{A} \in \mathbb{R}^{d \times d}$ be an arbitrary orthogonal matrix with $A_{ij} \neq 0$ for all $i = 1, 2, \dots, d$ and $j = 1, 2, \dots, d$. An infinite family of such matrices can be constructed using a Householder transformation: Choose an arbitrary $\alpha \in (0, 0.5)$ and consider the vector $\boldsymbol{v}$ with $v_1 = \sqrt{\alpha}$ and $v_i = \sqrt{\frac{1-\alpha}{d-1}}$ for $i = 2, 3, \dots, d$. By construction, we have $\boldsymbol{v}^T \boldsymbol{v} = 1$ and both $v_i \neq 0$ and $v_i \neq \sqrt{\frac{1}{2}}$ for all $i = 1, 2, \dots, d$. Define the matrix $\boldsymbol{A} = \boldsymbol{I}_d - 2\boldsymbol{v}\boldsymbol{v}^T$ and note that $A_{ii} = 1 - 2v_i^2 \neq 0$ for all $1, 2, \dots, d$ as well as $A_{ij} = -v_i v_j \neq 0$ for all $i \neq j$. Furthermore, $\boldsymbol{A}$ is orthogonal since

$$\boldsymbol{A}^T \boldsymbol{A} = \left(\boldsymbol{I}_d - 2\boldsymbol{v}\boldsymbol{v}^T\right)^T \left(\boldsymbol{I}_d - 2\boldsymbol{v}\boldsymbol{v}^T\right) = \boldsymbol{I}_d - 4\boldsymbol{v}\boldsymbol{v}^T + 4\boldsymbol{v}(\boldsymbol{v}^T\boldsymbol{v})\boldsymbol{v}^T = \boldsymbol{I}_d.$$

Since $\boldsymbol{A}$ is orthogonal, it is invertible and thus defines a bijective linear operator. The random variable $\boldsymbol{A}h(g(\mathbf{z})) \in \mathbb{R}^d$ is hence an independent, multivariate standard normal distribution since the covariance matrix $\boldsymbol{A}^T \boldsymbol{A}$ is equal to $\boldsymbol{I}_d$.

Since $h$ is bijective, it follows that $h^{-1}(\boldsymbol{A}h(g(\mathbf{z})))$ is an independent $d$-dimensional uniform distribution. Define the function $f : \operatorname{supp}(\mathbf{z}) \to \operatorname{supp}(\mathbf{z})$

$$f(\boldsymbol{u}) = g^{-1}(h^{-1}(\boldsymbol{A}h(g(\boldsymbol{u}))))$$

and note that by definition $f(\mathbf{z})$ has the same marginal distribution as $\mathbf{z}$ under $P$, $i.e.$, $P(\mathbf{z} \leq \boldsymbol{u}) = P(f(\mathbf{z}) \leq \boldsymbol{u})$ for all $\boldsymbol{u}$. Finally, for almost every $\boldsymbol{u} \in \operatorname{supp}(\mathbf{z})$, it holds that

$$\frac{\partial f_i(\boldsymbol{u})}{\partial u_j} = \frac{A_{ij} \cdot \frac{\partial h_j(g(\boldsymbol{u}))}{\partial v_j} \cdot \frac{\partial g_j(\boldsymbol{u})}{\partial u_j}}{\frac{\partial h_i(h_i^{-1}(\boldsymbol{A}h(g(\boldsymbol{u}))))}{\partial v_i} \cdot \frac{\partial g_i(g^{-1}(h^{-1}(\boldsymbol{A}h(g(\boldsymbol{u})))))}{\partial v_i}} \neq 0,$$

as claimed. Since the choice of $\boldsymbol{A}$ was arbitrary, there exists an infinite family of such functions $f$. $\qquad\square$

## C   Unsupervised learning of disentangled representations with VAEs

Variants of variational autoencoders [30] are considered the state-of-the-art for unsupervised disentanglement learning. One assumes a specific prior $P(\mathbf{z})$ on the latent space and then parameterizes the conditional probability $P(\mathbf{x}|\mathbf{z})$ with a deep neural network. Similarly, the distribution $P(\mathbf{z}|\mathbf{x})$ is approximated using a variational distribution $Q(\mathbf{z}|\mathbf{x})$, again parametrized using a deep neural network. One can then derive the following approximation to the maximum likelihood objective,

$$\max_{\phi, \theta} \quad \mathbb{E}_{p(\mathbf{x})}[\mathbb{E}_{q_\phi(\mathbf{z}|\mathbf{x})}[\log p_\theta(\mathbf{x}|\mathbf{z})] - D_{\mathrm{KL}}(q_\phi(\mathbf{z}|\mathbf{x})\|p(\mathbf{z}))] \tag{1}$$

which is also know as the evidence lower bound (ELBO). By carefully considering the KL term, one can encourage various properties of the resulting presentation. We will briefly review the main approaches.

**Bottleneck capacity.**   Higgins et al. [19] propose the $\beta$-VAE, introducing a hyperparameter in front of the KL regularizer of vanilla VAEs. They maximize the following expression:

$$\mathbb{E}_{p(\mathbf{x})}[\mathbb{E}_{q_\phi(\mathbf{z}|\mathbf{x})}[\log p_\theta(\mathbf{x}|\mathbf{z})] - \beta D_{\mathrm{KL}}(q_\phi(\mathbf{z}|\mathbf{x})\|p(\mathbf{z}))]$$

By setting $\beta > 1$, the encoder distribution will be forced to better match the factorized unit Gaussian prior. This procedure introduces additional constraints on the capacity of the latent bottleneck, encouraging the encoder to learn a disentangled representation for the data. Burgess et al. [6] argue that when the bottleneck has limited capacity, the network will be forced to specialize on the factor of variation that most contributes to a small reconstruction error. Therefore, they propose to progressively increase the bottleneck capacity, so that the encoder can focus on learning one factor of variation at the time:

$$\mathbb{E}_{p(\mathbf{x})}[\mathbb{E}_{q_\phi(\mathbf{z}|\mathbf{x})}[\log p_\theta(\mathbf{x}|\mathbf{z})] - \gamma|D_{\mathrm{KL}}(q_\phi(\mathbf{z}|\mathbf{x})\|p(\mathbf{z})) - C|]$$

where C is annealed from zero to some value which is large enough to produce good reconstruction. In the following, we refer to this model as AnnealedVAE.

**Penalizing the total correlation.** Let $I(\mathbf{x}; \mathbf{z})$ denote the mutual information between $\mathbf{x}$ and $\mathbf{z}$ and note that the second term in equation 1 can be rewritten as

$$\mathbb{E}_{p(\mathbf{x})}[D_{\mathrm{KL}}(q_\phi(\mathbf{z}|\mathbf{x})\|p(\mathbf{z}))] = I(\mathbf{x}; \mathbf{z}) + D_{\mathrm{KL}}(q(\mathbf{z})\|p(\mathbf{z})).$$

Therefore, when $\beta > 1$, $\beta$-VAE penalizes the mutual information between the latent representation and the data, thus constraining the capacity of the latent space. Furthermore, it pushes $q(\mathbf{z})$, the so called *aggregated posterior*, to match the prior and therefore to factorize, given a factorized prior. Kim & Mnih [29] argues that penalizing $I(\mathbf{x}; \mathbf{z})$ is neither necessary nor desirable for disentanglement. The FactorVAE [29] and the $\beta$-TCVAE [7] augment the VAE objective with an additional regularizer that specifically penalizes dependencies between the dimensions of the representation:

$$\mathbb{E}_{p(\mathbf{x})}[\mathbb{E}_{q_\phi(\mathbf{z}|\mathbf{x})}[\log p_\theta(\mathbf{x}|\mathbf{z})] - D_{\mathrm{KL}}(q_\phi(\mathbf{z}|\mathbf{x})\|p(\mathbf{z}))] - \gamma D_{\mathrm{KL}}(q(\mathbf{z})\|\prod_{j=1}^{d} q(\mathbf{z}_j)).$$

This last term is also known as *total correlation* [59]. The total correlation is intractable and vanilla Monte Carlo approximations require marginalization over the training set. [29] propose an estimate using the density ratio trick [43, 55] (FactorVAE). Samples from $\prod_{j=1}^{d} q(\mathbf{z}_j)$ can be obtained shuffling samples from $q(\mathbf{z})$ [1]. Concurrently, Chen et al. [7] propose a tractable biased Monte-Carlo estimate for the total correlation ($\beta$-TCVAE).

**Disentangled priors.** Kumar et al. [32] argue that a disentangled generative model requires a disentangled prior. This approach is related to the total correlation penalty, but now the aggregated posterior is pushed to match a factorized prior. Therefore

$$\mathbb{E}_{p(\mathbf{x})}[\mathbb{E}_{q_\phi(\mathbf{z}|\mathbf{x})}[\log p_\theta(\mathbf{x}|\mathbf{z})] - D_{\mathrm{KL}}(q_\phi(\mathbf{z}|\mathbf{x})\|p(\mathbf{z}))] - \lambda D(q(\mathbf{z})\|p(\mathbf{z})),$$

where $D$ is some (arbitrary) divergence. Since this term is intractable when $D$ is the KL divergence, they propose to match the moments of these distribution. In particular, they regularize the deviation of either $\mathrm{Cov}_{p(\mathbf{x})}[\mu_\phi(\mathbf{x})]$ or $\mathrm{Cov}_{q_\phi}[\mathbf{z}]$ from the identity matrix in the two variants of the DIP-VAE. This results in maximizing either the DIP-VAE-I objective

$$\mathbb{E}_{p(\mathbf{x})}[\mathbb{E}_{q_\phi(\mathbf{z}|\mathbf{x})}[\log p_\theta(\mathbf{x}|\mathbf{z})] - D_{\mathrm{KL}}(q_\phi(\mathbf{z}|\mathbf{x})\|p(\mathbf{z}))] - \lambda_{od} \sum_{i \neq j} \left[\mathrm{Cov}_{p(\mathbf{x})}[\mu_\phi(\mathbf{x})]\right]_{ij}^2 -$$

$$\lambda_d \sum_i \left(\left[\mathrm{Cov}_{p(\mathbf{x})}[\mu_\phi(\mathbf{x})]\right]_{ii} - 1\right)^2$$

or the DIP-VAE-II objective

$$\mathbb{E}_{p(\mathbf{x})}[\mathbb{E}_{q_\phi(\mathbf{z}|\mathbf{x})}[\log p_\theta(\mathbf{x}|\mathbf{z})] - D_{\mathrm{KL}}(q_\phi(\mathbf{z}|\mathbf{x})\|p(\mathbf{z}))] - \lambda_{od} \sum_{i \neq j} \left[\mathrm{Cov}_{q_\phi}[\mathbf{z}]\right]_{ij}^2 -$$

$$\lambda_d \sum_i \left(\left[\mathrm{Cov}_{q_\phi}[\mathbf{z}]\right]_{ii} - 1\right)^2.$$

# D  Implementation of metrics

All our metrics consider the expected representation of training samples (except total correlation for which we also consider the sampled representation as described in Section 4).

**BetaVAE metric.**   Higgins et al. [19] suggest to fix a random factor of variation in the underlying generative model and to sample two mini batches of observations $\mathbf{x}$. Disentanglement is then measured as the accuracy of a linear classifier that predicts the index of the fixed factor based on the coordinate-wise sum of absolute differences between the representation vectors in the two mini batches. We sample two batches of 64 points with a random factor fixed to a randomly sampled value across the two batches and the others varying randomly. We compute the mean representations for these points and take the absolute difference between pairs from the two batches. We then average these 64 values to form the features of a training (or testing) point. We train a Scikit-learn logistic regression with default parameters on 10 000 points. We test on 5000 points.

**FactorVAE metric**   Kim & Mnih [29] address several issues with this metric by using a majority vote classifier that predicts the index of the fixed ground-truth factor based on the index of the representation vector with the least variance. First, we estimate the variance of each latent dimension by embedding 10 000 random samples from the data set and we exclude collapsed dimensions with variance smaller than 0.05. Second, we generate the votes for the majority vote classifier by sampling a batch of 64 points, all with a factor fixed to the same random value. Third, we compute the variance of each dimension of their latent representation and divide by the variance of that dimension we computed on the data without interventions. The training point for the majority vote classifier consists of the index of the dimension with the smallest normalized variance. We train on 10 000 points and evaluate on 5000 points.

**Mutual Information Gap.**   Chen et al. [7] argue that the BetaVAE metric and the FactorVAE metric are neither general nor unbiased as they depend on some hyperparameters. They compute the mutual information between each ground truth factor and each dimension in the computed representation $r(\mathbf{x})$. For each ground-truth factor $\mathbf{z}_k$, they then consider the two dimensions in $r(\mathbf{x})$ that have the highest and second highest mutual information with $\mathbf{z}_k$. The *Mutual Information Gap (MIG)* is then defined as the average, normalized difference between the highest and second highest mutual information of each factor with the dimensions of the representation. The original metric was proposed evaluating the sampled representation. Instead, we consider the mean representation, in order to be consistent with the other metrics. We estimate the discrete mutual information by binning each dimension of the representations obtained from 10 000 points into 20 bins. Then, the score is computed as follows:

$$\frac{1}{K}\sum_{k=1}^{K}\frac{1}{H_{\mathbf{z}_k}}\left(I(\mathbf{v}_{j_k},\mathbf{z}_k)-\max_{j\neq j_k}I(\mathbf{v}_j,\mathbf{z}_k)\right)$$

Where $\mathbf{z}_k$ is a factor of variation, $\mathbf{v}_j$ is a dimension of the latent representation and $j_k = \arg\max_j I(\mathbf{v}_j, \mathbf{z}_k)$.

**Modularity.**   Ridgeway & Mozer [49] argue that two different properties of representations should be considered, i.e., *Modularity* and *Explicitness*. In a modular representation each dimension of $r(\mathbf{x})$ depends on at most a single factor of variation. In an explicit representation, the value of a factor of variation is easily predictable (i.e. with a linear model) from $r(\mathbf{x})$. They propose to measure the Modularity as the average normalized squared difference of the mutual information of the factor of variations with the highest and second-highest mutual information with a dimension of $r(\mathbf{x})$. They measure Explicitness as the ROC-AUC of a one-versus-rest logistic regression classifier trained to predict the factors of variation. In this study, we focus on Modularity as it is the property that corresponds to disentanglement. For the modularity score, we sample 10 000 points for which we obtain the latent representations. We discretize these points into 20 bins and compute the mutual information between representations and the values of the factors of variation. These values are stored in a matrix $\mathbf{m}$. For each dimension of the representation $i$, we compute a vector $\mathbf{t}_i$ as:

$$t_{i,f} = \begin{cases} \theta_i & \text{if } f = \arg\max_g m_{i,g} \\ 0 & \text{otherwise} \end{cases}$$

where $\theta_i = \max_g m_{ig}$. The modularity score is the average over the dimensions of the representation of $1 - \delta_i$ where:

$$\delta_i = \frac{\sum_f (m_{if} - t_{if})^2}{\theta_i^2 (N-1)}$$

and N is the number of factors.

**DCI Disentanglement.** Ridgeway & Mozer [49] consider three properties of representations, i.e., *Disentanglement*, *Completeness* and *Informativeness*. First, Eastwood & Williams [15] compute the importance of each dimension of the learned representation for predicting a factor of variation. The predictive importance of the dimensions of $r(\mathbf{x})$ can be computed with a Lasso or a Random Forest classifier. Disentanglement is the average of the difference from one of the entropy of the probability that a dimension of the learned representation is useful for predicting a factor weighted by the relative importance of each dimension. Completeness, is the average of the difference from one of the entropy of the probability that a factor of variation is captured by a dimension of the learned representation. Finally, the Informativeness can be computed as the prediction error of predicting the factors of variations. We sample $10\,000$ and $5000$ training and test points respectively. For each factor, we fit gradient boosted trees from Scikit-learn with the default setting. From this model, we extract the importance weights for the feature dimensions. We take the absolute value of these weights and use them to form the importance matrix $R$, whose rows correspond to factors and columns to the representation. To compute the disentanglement score, we first subtract from 1 the entropy of each column of this matrix (we treat the columns as a distribution by normalizing them). This gives a vector of length equal to the dimensionality of the latent space. Then, we compute the relative importance of each dimension by $\rho_i = \sum_j R_{ij} / \sum_{ij} R_{ij}$ and the disentanglement score as $\sum_i \rho_i (1 - H(R_i))$.

**SAP score.** Kumar et al. [32] propose to compute the $R^2$ score of the linear regression predicting the factor values from each dimension of the learned representation. For discrete factors, they propose to train a classifier. The *Separated Attribute Predictability (SAP)* score is the average difference of the prediction error of the two most predictive latent dimensions for each factor. We sample $10\,000$ points for training and $5000$ for testing. We then compute a score matrix containing the prediction error on the test set for a linear SVM with $C = 0.01$ predicting the value of a factor from a single latent dimension. The SAP score is computed as the average across factors of the difference between the top two most predictive latent dimensions.

**Downstream task.** We sample training sets of different sizes: 10, 100, 1000 and $10\,000$ points. We always evaluate on 5000 samples. We consider as a downstream task the prediction of the values of each factor from $r(\mathbf{x})$. For each factor we fit a different model and report then report the average test accuracy across factors. We consider two different models. First, we train a cross validated logistic regression from Scikit-learn with 10 different values for the regularization strength ($Cs = 10$) and 5 folds. Finally, we train a gradient boosting classifier from Scikit-learn with default parameters.

**Total correlation based on fitted Gaussian.** We sample $10\,000$ points and obtain their latent representation $r(\mathbf{x})$ by either sampling from the encoder distribution of by taking its mean. We then compute the mean $\mu_{r(\mathbf{x})}$ and covariance matrix $\Sigma_{r(\mathbf{x})}$ of these points and compute the total correlation of a Gaussian with mean $\mu_{r(\mathbf{x})}$ and covariance matrix $\Sigma_{r(\mathbf{x})}$, i.e.,

$$D_{\mathrm{KL}} \left( \mathcal{N}(\mu_{r(\mathbf{x})}, \Sigma_{r(\mathbf{x})}) \Big\| \prod_j \mathcal{N}(\mu_{r(\mathbf{x})_j}, \Sigma_{r(\mathbf{x})_{jj}}) \right)$$

where $j$ indexes the dimensions in the latent space. We choose this approach for the following reasons: In this study, we compute statistics of $r(\mathbf{x})$ which can be either sampled from the probabilistic encoder or taken to be its mean. We argue that estimating the total correlation as in [29] is not suitable for this comparison as it consistently underestimate the true value (see Figure 7 in [29]) and depends on a non-convex optimization procedure (for fitting the discriminator). The estimate of [7] is also not suitable as the mean representation is a deterministic function for the data, therefore we cannot use the encoder distribution for the estimate. Furthermore, we argue that the total correlation based on the fitted Gaussian provides a simple and robust way to detect if a representation is not factorizing based

on the first two moments. In particular, if it is high, it is a strong signal that the representation is not factorizing (while a low score may not imply the opposite). We note that this procedure is similar to the penalty of DIP-VAE-I. Therefore, it is not surprising that DIP-VAE-I achieves a low score for the mean representation.

## E  Experimental conditions and guiding principles.

In our study, we seek controlled, fair and reproducible experimental conditions. We consider the case in which we can sample from a well defined and known ground-truth generative model by first sampling the factors of variations from a distribution $P(\mathbf{z})$ and then sampling an observation from $P(\mathbf{x}|\mathbf{z})$. Our experimental protocol works as follows: During training, we only observe the samples of $\mathbf{x}$ obtained by marginalizing $P(\mathbf{x}|\mathbf{z})$ over $P(\mathbf{z})$. After training, we obtain a representation $r(\mathbf{x})$ by either taking a sample from the probabilistic encoder $Q(\mathbf{z}|\mathbf{x})$ or by taking its mean. Typically, disentanglement metrics consider the latter as the representation $r(\mathbf{x})$. During the evaluation, we assume to have access to the whole generative model, i.e. we can draw samples from both $P(\mathbf{z})$ and $P(\mathbf{x}|\mathbf{z})$. In this way, we can perform interventions on the latent factors as required by certain evaluation metrics. We explicitly note that we effectively consider the statistical learning problem where we optimize the loss and the metrics on the known data generating distribution. As a result, we do not use separate train and test sets but always take i.i.d. samples from the known ground-truth distribution. This is justified as the statistical problem is well defined and it allows us to remove the additional complexity of dealing with overfitting and empirical risk minimization.

## F  Limitations of our study.

While we aim to provide a useful and fair experimental study, there are clear limitations to the conclusions that can be drawn from it due to design choices that we have taken. In all these choices, we have aimed to capture what is considered the state-of-the-art inductive bias in the community.

On the data set side, we only consider images with a heavy focus on synthetic images. We do not explore other modalities and we only consider the toy scenario in which we have access to a data generative process with uniformly distributed factors of variations. Furthermore, all our data sets have a small number of independent discrete factors of variations without any confounding variables.

For the methods, we only consider the inductive bias of convolutional architectures. We do not test fully connected architectures or additional techniques such as skip connections. Furthermore, we do not explore different activation functions, reconstruction losses or different number of layers. We also do not vary any other hyperparameters other than the regularization weight. In particular, we do not evaluate the role of different latent space sizes, optimizers and batch sizes. We do not test the sample efficiency of the metrics but simply set the size of the train and test set to large values.

Implementing the different disentanglement methods and metrics has proven to be a difficult endeavour. Few "official" open source implementations are available and there are many small details to consider. We take a best-effort approach to these implementations and implemented all the methods and metrics from scratch as any sound machine learning practitioner might do based on the original papers. When taking different implementation choices than the original papers, we explicitly state and motivate them.

**Inductive biases.** To fairly evaluate the different approaches, we separate the effect of regularization (in the form of model choice and regularization strength) from the other inductive biases (e.g., the choice of the neural architecture). Each method uses the same convolutional architecture, optimizer, hyperparameters of the optimizer and batch size. All methods use a Gaussian encoder where the mean and the log variance of each latent factor is parametrized by the deep neural network, a Bernoulli decoder and latent dimension fixed to 10. We note that these are all standard choices in prior work [19, 29].

We choose six different regularization strength, i.e., hyperparameter values, for each of the considered methods. The key idea was to take a wide enough set to ensure that there are useful hyperparameters for different settings for each method and not to focus on specific values known to work for specific data sets. However, the values are partially based on the ranges that are prescribed in the literature (including the hyperparameters suggested by the authors).

Table 2: Encoder and Decoder architecture for the main experiment.

| Encoder | Decoder |
|---|---|
| Input: $64 \times 64 \times$ number of channels | Input: $\mathbb{R}^{10}$ |
| $4 \times 4$ conv, 32 ReLU, stride 2 | FC, 256 ReLU |
| $4 \times 4$ conv, 32 ReLU, stride 2 | FC, $4 \times 4 \times 64$ ReLU |
| $4 \times 4$ conv, 64 ReLU, stride 2 | $4 \times 4$ upconv, 64 ReLU, stride 2 |
| $4 \times 4$ conv, 64 ReLU, stride 2 | $4 \times 4$ upconv, 32 ReLU, stride 2 |
| FC 256, F2 $2 \times 10$ | $4 \times 4$ upconv, 32 ReLU, stride 2 |
| | $4 \times 4$ upconv, number of channels, stride 2 |

## G    Differences with previous implementations.

As described above, we use a single choice of architecture, batch size and optimizer for all the methods which might deviate from the settings considered in the original papers. However, we argue that unification of these choices is the only way to guarantee a fair comparison among the different methods such that valid conclusions may be drawn in between methods. The largest change is that for DIP-VAE and for $\beta$-TCVAE we used a batch size of 64 instead of 400 and 2048 respectively. However, Chen et al. [7] shows in Section H.2 of the Appendix that the bias in the mini-batch estimation of the total correlation does not significantly affect the performances of their model even with small batch sizes. For DIP-VAE-II, we did not implement the additional regularizer on the third order central moments since no implementation details are provided and since this regularizer is only used on specific data sets.

Our implementations of the disentanglement metrics deviate from the implementations in the original papers as follows: First, we strictly enforce that all factors of variations are treated as discrete variables as this corresponds to the assumed ground-truth model in all our data sets. Hence, we used classification instead of regression for the SAP score and the disentanglement score of [15]. This is important as it does not make sense to use regression on true factors of variations that are discrete (e.g., shape on dSprites). Second, wherever possible, we resorted to using the default, well-tested Scikit-learn [45] implementations instead of using custom implementations with potentially hard to set hyperparameters. Third, for the Mutual Information Gap [7], we estimate the *discrete* mutual information (as opposed to continuous) on the *mean* representation (as opposed to sampled) on a *subset* of the samples (as opposed to the whole data set). We argue that this is the correct choice as the mean is usually taken to be the representation. Hence, it would be wrong to consider the full Gaussian encoder or samples thereof as that would correspond to a different representation. Finally, we fix the number of sampled train and test points across all metrics to a large value to ensure robustness.

## H    Main experiment hyperparameters

In our study, we fix all hyperparameters except one per each model. Model specific hyperparameters can be found in Table 3. The common architecture is depicted in Table 2 along with the other fixed hyperparameters in Table 4a. For the discriminator in FactorVAE we use the architecture in Table 4b with hyperparameters in Table 4c. All the hyperparameters for which we report single values were not varied and are selected based on the literature.

## I    Data sets and preprocessing

All the data sets contains images with pixels between 0 and 1. **Color-dSprites:** Every time we sample a point, we also sample a random scaling for each channel uniformly between 0.5 and 1. **Noisy-dSprites:** Every time we sample a point, we fill the background with uniform noise. **Scream-dSprites:** Every time we sample a point, we sample a random $64 \times 64$ patch of *The Scream* painting. We then change the color distribution by adding a random uniform number to each channel and divide the result by two. Then, we embed the dSprites shape by inverting the colors of each of its pixels.

Table 3: Model's hyperparameters. We allow a sweep over a single hyperparameter for each model.

| Model | Parameter | Values |
|---|---|---|
| $\beta$-VAE | $\beta$ | [1, 2, 4, 6, 8, 16] |
| AnnealedVAE | $c_{max}$ | [5, 10, 25, 50, 75, 100] |
| | iteration threshold | 100000 |
| | $\gamma$ | 1000 |
| FactorVAE | $\gamma$ | [10, 20, 30, 40, 50, 100] |
| DIP-VAE-I | $\lambda_{od}$ | [1, 2, 5, 10, 20, 50] |
| | $\lambda_d$ | $10\lambda_{od}$ |
| DIP-VAE-II | $\lambda_{od}$ | [1, 2, 5, 10, 20, 50] |
| | $\lambda_d$ | $\lambda_{od}$ |
| $\beta$-TCVAE | $\beta$ | [1, 2, 4, 6, 8, 10] |

Table 4: Other fixed hyperparameters.

(a) Hyperparameters common to each of the considered methods.

| Parameter | Values |
|---|---|
| Batch size | 64 |
| Latent space dimension | 10 |
| Optimizer | Adam |
| Adam: beta1 | 0.9 |
| Adam: beta2 | 0.999 |
| Adam: epsilon | 1e-8 |
| Adam: learning rate | 0.0001 |
| Decoder type | Bernoulli |
| Training steps | 300000 |

(b) Architecture for the discriminator in FactorVAE.

| Discriminator |
|---|
| FC, 1000 leaky ReLU |
| FC, 1000 leaky ReLU |
| FC, 1000 leaky ReLU |
| FC, 1000 leaky ReLU |
| FC, 1000 leaky ReLU |
| FC, 1000 leaky ReLU |
| FC, 2 |

(c) Parameters for the discriminator in FactorVAE.

| Parameter | Values |
|---|---|
| Batch size | 64 |
| Optimizer | Adam |
| Adam: beta1 | 0.5 |
| Adam: beta2 | 0.9 |
| Adam: epsilon | 1e-8 |
| Adam: learning rate | 0.0001 |

# J    Detailed experimental results

Given the breadth of the experimental study, we summarized our key findings in Section 4 and presented figures that we picked to be representative of our results. This section contains a self-contained presentation of all our experimental results. In particular, we present a complete set of plots for the different methods, data sets and disentanglement metrics.

## J.1    Can one achieve a good reconstruction error across data sets and models?

First, we check for each data set that we manage to train models that achieve reasonable reconstructions. Therefore, for each data set we sample a random model and show real samples next to their reconstructions. The results are depicted in Figure 7. As expected, the additional variants of dSprites with continuous noise variables are harder than the original data set. On Noisy-dSprites and Color-dSprites the models produce reasonable reconstructions with the noise on Noisy-dSprites being ignored. Scream-dSprites is even harder and we observe that the shape information is lost. On the other data sets, we observe that reconstructions are blurry but objects are distinguishable. SmallNORB seems to be the most challenging data set.

## J.2    Can current methods enforce a uncorrelated aggregated posterior and representation?

We investigate whether the considered unsupervised disentanglement approaches are effective at enforcing a factorizing and thus uncorrelated aggregated posterior. For each trained model, we sample 10 000 images and compute a sample from the corresponding approximate posterior. We then fit a multivariate Gaussian distribution over these 10 000 samples by computing the empirical mean and covariance matrix. Finally, we compute the total correlation of the fitted Gaussian and report the median value for each data set, method and hyperparameter value.

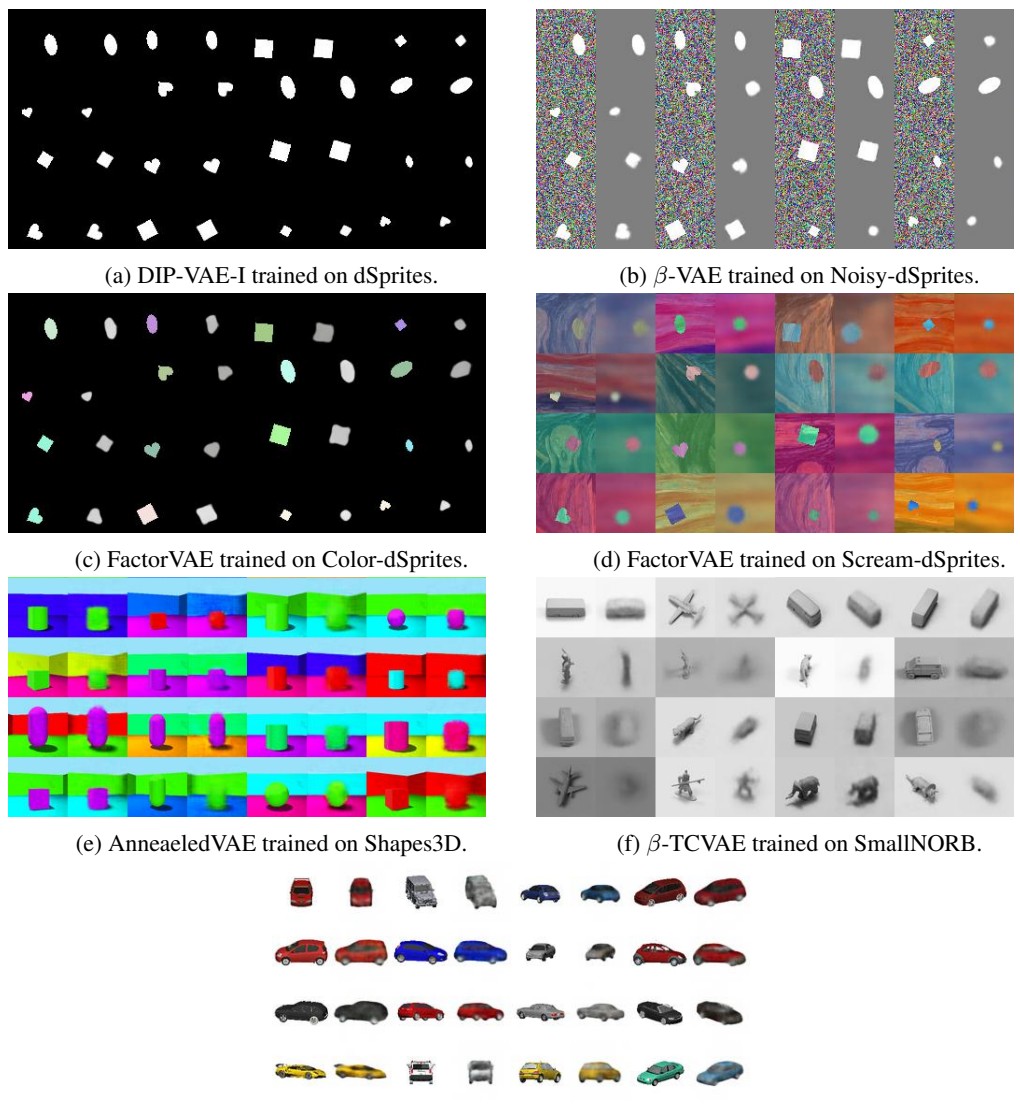

(a) DIP-VAE-I trained on dSprites.

(b) $\beta$-VAE trained on Noisy-dSprites.

(c) FactorVAE trained on Color-dSprites.

(d) FactorVAE trained on Scream-dSprites.

(e) AnneaeledVAE trained on Shapes3D.

(f) $\beta$-TCVAE trained on SmallNORB.

(g) Reconstructions for a DIP-VAE-II trained on Cars3D.

Figure 7: Reconstructions for different data sets and methods. Odd columns show real samples and even columns their reconstruction. As expected, the additional variants of dSprites with continuous noise variables are harder than the original data set. On Noisy-dSprites and Color-dSprites the models produce reasonable reconstructions with the noise on Noisy-dSprites being ignored. Scream-dSprites is even harder and we observe that the shape information is lost. On the other data sets, we observe that reconstructions are blurry but objects are distinguishable.

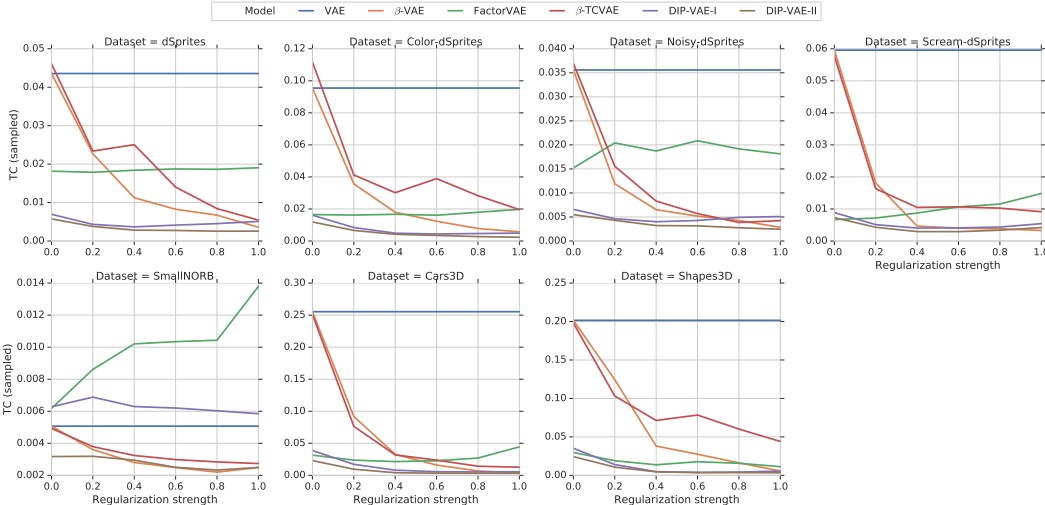

Figure 8: Total correlation of sampled representation plotted against regularization strength for different data sets and approaches (except AnnealedVAE). The total correlation of the sampled representation decreases as the regularization strength is increased.

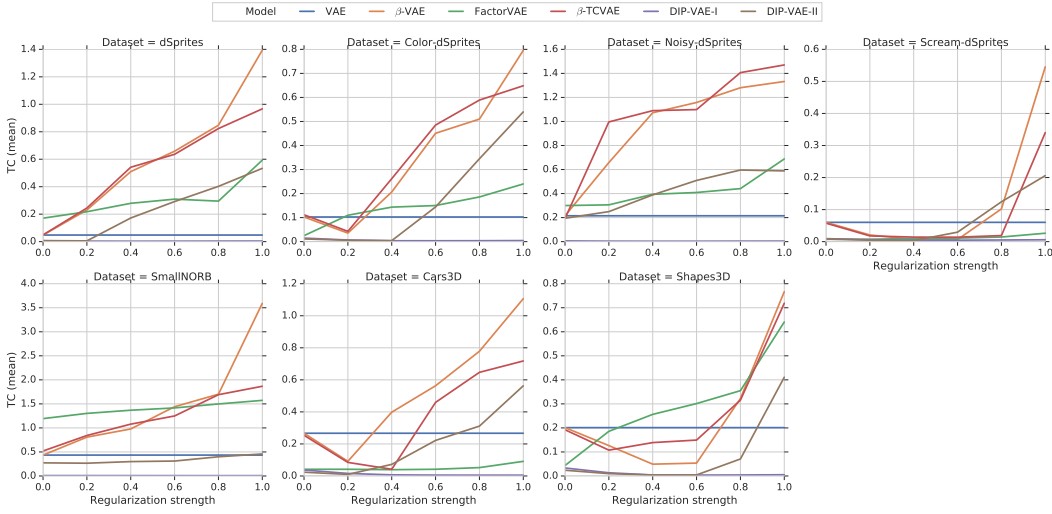

Figure 9: Total correlation of mean representation plotted against regularization strength for different data sets and approaches (except AnnealedVAE). The total correlation of the mean representation does not necessarily decrease as the regularization strength is increased.

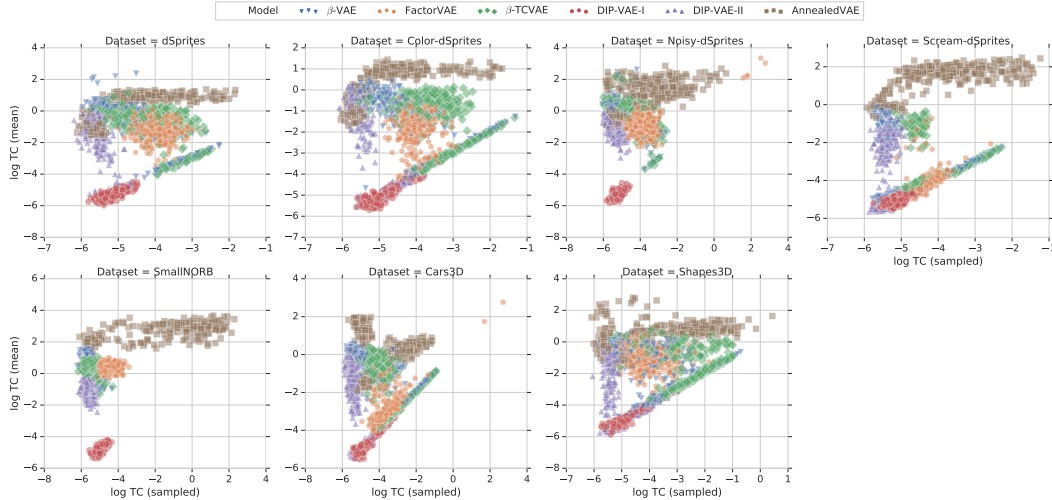

Figure 10: Log total correlation of mean vs sampled representations. For a large number of models, the total correlation of the mean representation is higher than that of the sampled representation.

Figure 8 shows the total correlation of the sampled representation plotted against the regularization strength for each data set and method except AnnealedVAE. On all data sets except SmallNORB, we observe that plain vanilla variational autoencoders (i.e. the $\beta$-VAE model with $\beta = 1$) exhibit the highest total correlation. For $\beta$-VAE and $\beta$-TCVAE, it can be clearly seen that the total correlation of the sampled representation decreases on all data sets as the regularization strength (in the form of $\beta$) is increased. The two variants of DIP-VAE exhibit low total correlation across the data sets except DIP-VAE-I which incurs a slightly higher total correlation on SmallNORB compared to a vanilla VAE. Increased regularization in the DIP-VAE objective also seems to lead a reduced total correlation, even if the effect is not as pronounced as for $\beta$-VAE and $\beta$-TCVAE. While FactorVAE achieves a low total correlation on all data sets except on SmallNORB, we observe that the total correlation does not seem to decrease with increasing regularization strength. We further observe that AnnealedVAE (shown in Figure 24) is much more sensitive to the regularization strength. However, on all data sets except Scream-dSprites (on which AnnealedVAE performs poorly), the total correlation seems to decrease with increased regularization strength.

While many of the considered methods aim to enforce a factorizing aggregated posterior, they use the mean vector of the Gaussian encoder as the representation and not a sample from the Gaussian encoder. This may seem like a minor, irrelevant modification; however, it is not clear whether a factorizing aggregated posterior also ensures that the dimensions of the mean representation are uncorrelated. To test whether this is true, we compute the mean of the Gaussian encoder for the same 10 000 samples, fit a multivariate Gaussian and compute the total correlation of that fitted Gaussian. Figure 9 shows the total correlation of the mean representation plotted against the regularization strength for each data set and method except AnnealedVAE. We observe that, for $\beta$-VAE and $\beta$-TCVAE, increased regularization leads to a substantially increased total correlation of the mean representations. This effect can also be observed for for FactorVAE, albeit in a less extreme fashion. For DIP-VAE-I, we observe that the total correlation of the mean representation is consistently low. This is not surprising as the DIP-VAE-I objective directly optimizes the covariance matrix of the mean representation to be diagonal which implies that the corresponding total correlation (as we compute it) is low. The DIP-VAE-II objective which enforces the covariance matrix of the sampled representation to be diagonal seems to lead to a factorized mean representation on some data sets (for example Shapes3D and Cars3D), but also seems to fail on others (dSprites). For AnnealedVAE (shown in Figure 25), we overall observe mean representations with a very high total correlation. In Figure 10, we further plot the log total correlations of the sampled representations versus the mean representations for each of the trained models. It can be clearly seen that for a large number of models, the total correlation of the mean representations is much higher than that of the sampled representations. The same trend can be seen computing the average discrete mutual information of the representation. In this case, the DIP-VAE-I exhibit increasing mutual information in both the

mean and sampled representation. This is to be expected as DIP-VAE-I enforces a variance of one for the mean representation.

**Implications.** Overall, these results lead us to conclude with minor exceptions that the considered methods are effective at enforcing an aggregated posterior whose individual dimensions are not correlated but that this does not seem to imply that the dimensions of the mean representation (usually used for representation) are uncorrelated.

### J.3 How much do existing disentanglement metrics agree?

As there exists no single, common definition of disentanglement, an interesting question is to see how much different proposed metrics agree. Figure 11 shows pairwise scatter plots of the different considered metrics on dSprites where each point corresponds to a trained model, while Figure 12 shows the Spearman rank correlation between different disentanglement metrics on different data sets. Overall, we observe that all metrics except Modularity seem to be correlated strongly on the data sets dSprites, Color-dSprites and Scream-dSprites and mildly on the other data sets. There appear to be two pairs among these metrics that capture particularly similar notions: the BetaVAE and the FactorVAE score as well as the Mutual Information Gap and DCI Disentanglement.

**Implication.** All disentanglement metrics except Modularity appear to be correlated. However, the level of correlation changes between different data sets.

### J.4 How important are different models and hyperparameters for disentanglement?

The primary motivation behind the considered methods is that they should lead to improved disentanglement scores. This raises the question how disentanglement is affected by the model choice, the hyperparameter selection and randomness (in the form of different random seeds). To investigate this, we compute all the considered disentanglement metrics for each of our trained models. In Figure 13, we show the range of attainable disentanglement scores for each method on each data set. We observe that these ranges are heavily overlapping for different models leading us to (qualitatively) conclude that the choice of hyperparameters and the random seed seems to be substantially more important than the choice of objective function. While certain models seem to attain better maximum scores on specific data sets and disentanglement metrics, we do not observe any consistent pattern that one model is consistently better than the other. Furthermore, we note that in our study we have fixed the range of hyperparameters a priori to six different values for each model and did not explore additional hyperparameters based on the results (as that would bias our study). However, this also means that specific models may have performed better than in Figure 13 if we had chosen a different set of hyperparameters. In Figure 14, we further show the impact of randomness in the form of random seeds on the disentanglement scores. Each violin plot shows the distribution of the disentanglement metric across all 50 trained models for each model and hyperparameter setting on Cars3D. We clearly see that randomness (in the form of different random seeds) has a substantial impact on the attained result and that a good run with a bad hyperparameter can beat a bad run with a good hyperparameter in many cases.

Finally, we perform a variance analysis by trying to predict the different disentanglement scores using ordinary least squares for each data set: If we allow the score to depend only on the objective function (categorical variable), we are only able to explain $37\%$ of the variance of the scores on average. Similarly, if the score depends on the Cartesian product of objective function and regularization strength (again categorical), we are able to explain $59\%$ of the variance while the rest is due to the random seed. In Table 5, we report the percentage of variance explained for the different metrics in each data set considering the regularization strength or not.

**Implication.** The disentanglement scores of unsupervised models are heavily influenced by randomness (in the form of the random seed) and the choice of the hyperparameter (in the form of the regularization strength). The objective function appears to have less impact.

### J.5 Are there reliable recipes for model selection?

In this section, we investigate how good hyperparameters can be chosen and how we can distinguish between good and bad training runs. In this paper, we advocate that model selection *should not*

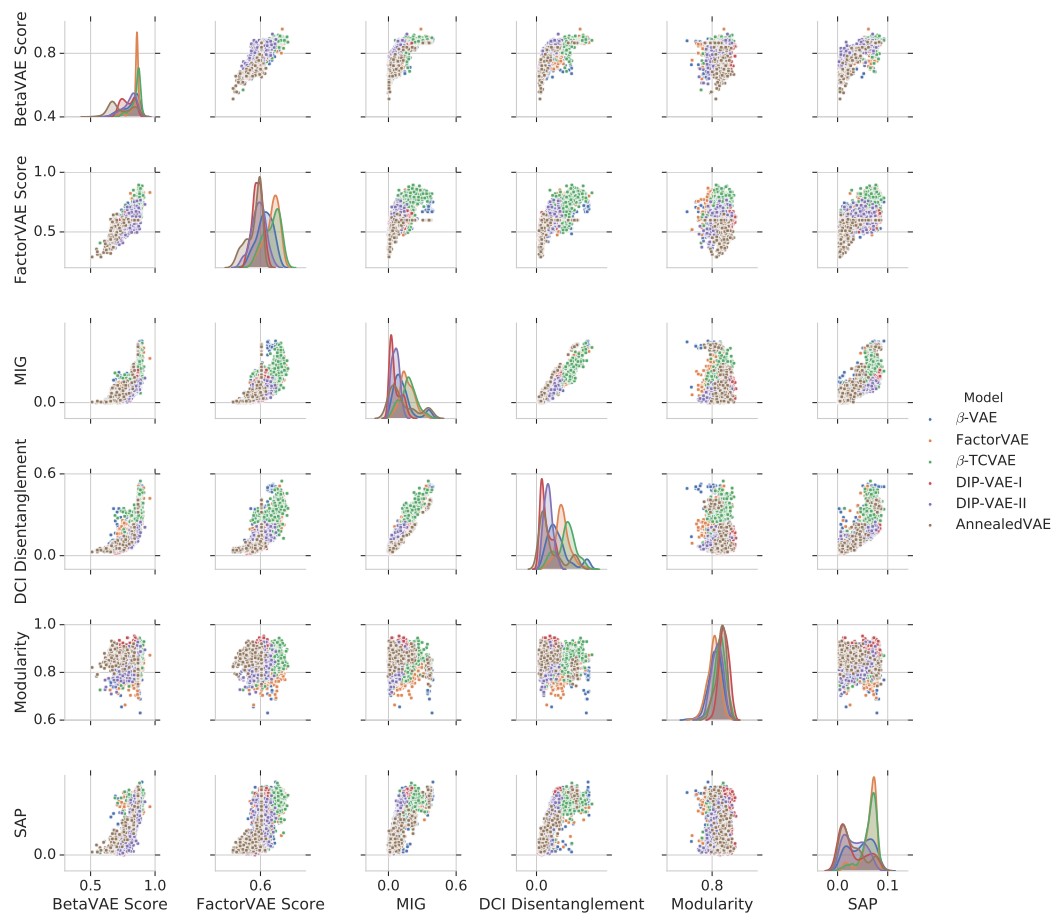

Figure 11: Pairwise scatter plots of different disentanglement metrics on dSprites. All the metrics except Modularity appear to be correlated. The strongest correlation seems to be between MIG and DCI Disentanglement.

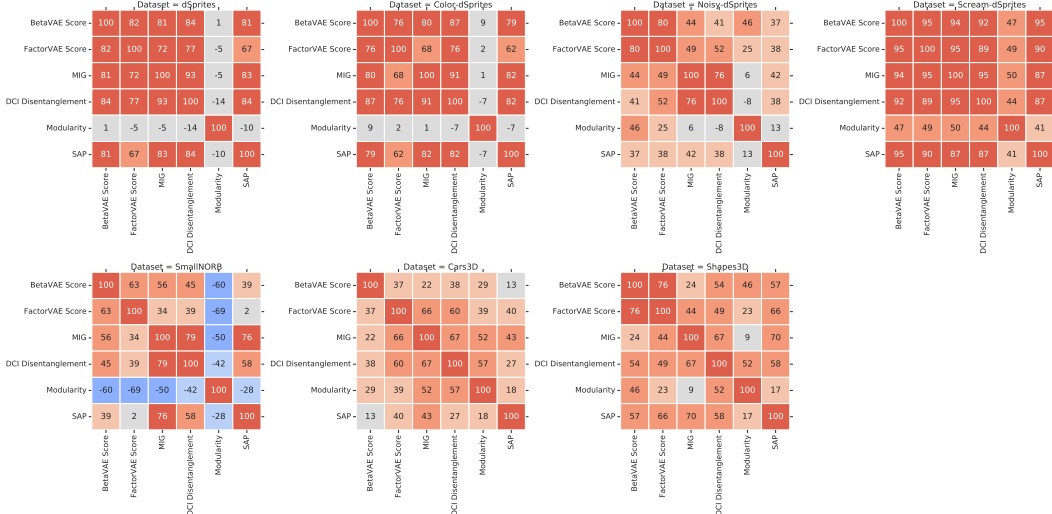

Figure 12: Rank correlation of different metrics on different data sets. Overall, we observe that all metrics except Modularity seem to be strongly correlated on the data sets dSprites, Color-dSprites and Scream-dSprites and mildly on the other data sets. There appear to be two pairs among these metrics that capture particularly similar notions: the BetaVAE and the FactorVAE score as well as the Mutual Information Gap and DCI Disentanglement.

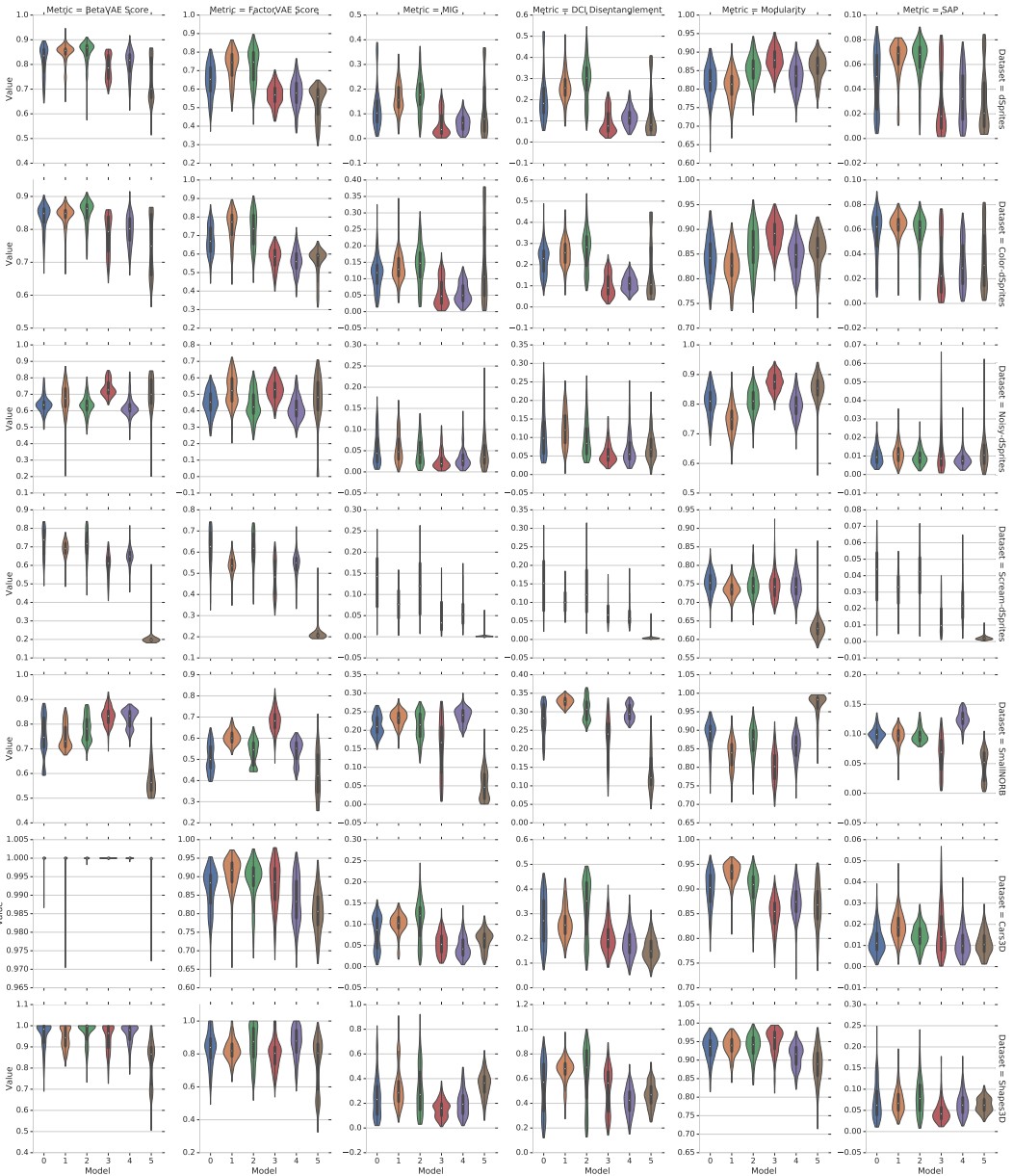

Figure 13: Score for each method for each score (column) and data set (row). Models are abbreviated (0=$\beta$-VAE, 1=FactorVAE, 2=$\beta$-TCVAE, 3=DIP-VAE-I, 4=DIP-VAE-II, 5=AnnealedVAE). The scores are heavily overlapping and we do not observe a consistent pattern. We conclude that hyperparameters matter more than the model choice.

Table 5: Variance of the disentanglement scores explained by the objective function or its cartesian product with the hyperparameters. The variance explained is computed regressing using ordinary least squares.

(a) Percentage of variance explained regressing the disentanglement scores on the different data sets from the objective function only. We abbreviate the DCI Disentanglement as "DCI Dis.".

|  | BetaVAE Score | DCI Disentanglement | FactorVAE Score | MIG | Modularity | SAP |
|---|---|---|---|---|---|---|
| Cars3D | 1% | 36% | 26% | 34% | 37% | 13% |
| Color-dSprites | 30% | 39% | 52% | 26% | 23% | 29% |
| Noisy-dSprites | 17% | 21% | 17% | 11% | 41% | 6% |
| Scream-dSprites | 89% | 50% | 76% | 45% | 60% | 56% |
| Shapes3D | 31% | 21% | 14% | 20% | 26% | 10% |
| SmallNORB | 68% | 71% | 58% | 71% | 62% | 62% |
| dSprites | 29% | 41% | 47% | 26% | 29% | 31% |

(b) Percentage of variance explained regressing the disentanglement scores on the different data sets from the Cartesian product of objective function and regularization strength. We abbreviate the DCI Disentanglement as "DCI Dis.".

|  | BetaVAE Score | DCI Disentanglement | FactorVAE Score | MIG | Modularity | SAP |
|---|---|---|---|---|---|---|
| Cars3D | 4% | 69% | 42% | 59% | 51% | 17% |
| Color-dSprites | 69% | 80% | 61% | 76% | 40% | 56% |
| Noisy-dSprites | 26% | 42% | 25% | 29% | 50% | 20% |
| Scream-dSprites | 93% | 74% | 83% | 66% | 68% | 75% |
| Shapes3D | 61% | 78% | 53% | 59% | 49% | 35% |
| SmallNORB | 87% | 89% | 81% | 88% | 72% | 82% |
| dSprites | 59% | 77% | 54% | 72% | 39% | 56% |

depend on the considered disentanglement score for the following reasons: The point of unsupervised learning of disentangled representation is that there is no access to the labels as otherwise we could incorporate them and would have to compare to semi-supervised and fully supervised methods. All the disentanglement metrics considered in this paper require a substantial amount of ground-truth labels or the full generative model (for example for the BetaVAE and the FactorVAE metric). Hence, one may substantially bias the results of a study by tuning hyperparameters based on (supervised) disentanglement metrics. Furthermore, we argue that it is not sufficient to fix a set of hyperparameters *a priori* and then show that one of those hyperparameters and a specific random seed achieves a good disentanglement score as it amounts to showing the existence of a good model, but does not guide the practitioner in finding it. Finally, in many practical settings, we might not even have access to adequate labels as it may be hard to identify the true underlying factor of variations, in particular, if we consider data modalities that are less suitable to human interpretation than images.

In the remainder of this section, we hence investigate and assess different ways how hyperparameters and good model runs could be chosen. In this study, we focus on choosing the learning model and the regularization strength corresponding to that loss function. However, we note that in practice this problem is likely even harder as a practitioner might also want to tune other modeling choices such architecture or optimizer.

**General recipes for hyperparameter selection.**    We first investigate whether we may find generally applicable "rules of thumb" for choosing the hyperparameters. For this, we plot in Figure 15 different disentanglement metrics against different regularization strengths for each model and each data set. The values correspond to the median obtained values across 50 random seeds for each model, hyperparameter and data set. There seems to be no model dominating all the others and for each model there does not seem to be a consistent strategy in choosing the regularization strength to maximize disentanglement scores. Furthermore, even if we could identify a good objective function and corresponding hyperparameter value, we still could not distinguish between a good and a bad training run.

Table 6: Probability of outperforming random model selection on a different random seed. A random disentanglement metric and data set is sampled and used for model selection. That model is then compared to a randomly selected model: (i) on the same metric and data set, (ii) on the same metric and a random different data set, (iii) on a random different metric and the same data set, and (iv) on a random different metric and a random different data set. The results are averaged across 10 000 random draws.

|  | Random different data set | Same data set |
| --- | --- | --- |
| Random different metric | 54.9% | 62.6% |
| Same metric | 59.3% | 80.7% |

**Model selection based on unsupervised scores.** Another approach could be to select hyperparameters based on unsupervised scores such as the reconstruction error, the KL divergence between the prior and the approximate posterior, the Evidence Lower Bound or the estimated total correlation of the sampled representation. This would have the advantage that we could select specific trained models and not just good hyperparameter settings whose median trained model would perform well. To test whether such an approach is fruitful, we compute the rank correlation between these unsupervised metrics and the disentanglement metrics and present it in Figure 16. While we do observe some correlations, no clear pattern emerges which leads us to conclude that this approach is unlikely to be successful in practice.

**Hyperparameter selection based on transfer.** The final strategy for hyperparameter selection that we consider is based on transferring good settings across data sets. The key idea is that good hyperparameter settings may be inferred on data sets where we have labels available (such as dSprites) and then applied to novel data sets. To test this idea, we plot in Figure 18 the different disentanglement scores obtained on dSprites against the scores obtained on other data sets. To ensure robustness of the results, we again consider the median across all 50 runs for each model, regularization strength, and data set. We observe that the scores on Color-dSprites seem to be strongly correlated with the scores obtained on the regular version of dSprites. Figure 17 further shows the rank correlations obtained between different data sets for each disentanglement scores. This confirms the strong and consistent correlation between dSprites and Color-dSprites. While these result suggest that some transfer of hyperparameters is possible, it does not allow us to distinguish between good and bad random seeds on the target data set.

To illustrate this, we compare such a transfer based approach to hyperparameter selection to random model selection as follows: We first randomly sample one of our 50 random seeds and consider the set of trained models with that random seed. First, we sample one of our 50 random seeds, a random disentanglement metric and a data set and use them to select the hyperparameter setting with the highest attained score. Then, we compare that selected hyperparameter setting to a randomly selected model on either the same or a random different data set, based on either the same or a random different metric and for a randomly sampled seed. Finally, we report the percentage of trials in which this transfer strategy outperforms or performs equally well as random model selection across 10 000 trials in Table 6. If we choose the same metric and the same data set (but a different random seed), we obtain a score of 80.7%. If we aim to transfer for the same metric across data sets, we achieve around 59.3%. Finally, if we transfer both across metrics and data sets, our performance drops to 54.9%.

**Implications.** Unsupervised model selection remains an unsolved problem. Transfer of good hyperparameters between metrics and data sets does not seem to work as there appears to be no unsupervised way to distinguish between good and bad random seeds on the target task.

### J.6 Are these disentangled representations useful for downstream tasks in terms of the sample complexity of learning?

One of the key motivations behind disentangled representations is that they are assumed to be useful for later downstream tasks. In particular, it is argued that disentanglement should lead to a better sample complexity of learning [4, 52, 46]. In this section, we consider the simplest downstream classification task where the goal is to recover the true factors of variations from

the learned representation using either multi-class logistic regression (LR) or gradient boosted trees (GBT). Our goal is to investigate the relationship between disentanglement and the average classification accuracy on these downstream tasks as well as whether better disentanglement leads to a decreased sample complexity of learning.

To compute the classification accuracy for each trained model, we sample true factors of variations and observations from our ground truth generative models. We then feed the observations into our trained model and take the mean of the Gaussian encoder as the representations. Finally, we predict each of the ground-truth factors based on the representations with a separate learning algorithm. We consider both a 5-fold cross-validated multi-class logistic regression as well as gradient boosted trees of the Scikit-learn package. For each of these methods, we train on 10, 100, 1000 and 10 000 samples. We compute the average accuracy across all factors of variation using an additional set 10 000 randomly drawn samples.

Figure 19 shows the rank correlations between the disentanglement metrics and the downstream performance for all considered data sets. We observe that all metrics except Modularity seem to be correlated with increased downstream performance on the different variations of dSprites and to some degree on Shapes3D. However, it is not clear whether this is due to the fact that disentangled representations perform better or whether some of these scores actually also (partially) capture the informativeness of the evaluated representation. Furthermore, the correlation is weaker or inexistent on other data sets (e.g., Cars3D).

To assess the sample complexity argument we compute for each trained model a statistical efficiency score which we define as the average accuracy based on 100 samples divided by the average accuracy based on 10 000 samples for either the logistic regression or the gradient boosted trees. The key idea is that if disentangled representations lead to sample efficiency, then they should also exhibit a higher statistical efficiency score. The corresponding results are shown in Figures 20 and 21 where we plot the statistical efficiency versus different disentanglement metrics for different data sets and models and in Figure 19 where we show rank correlations. Overall, we do not observe conclusive evidence that models with higher disentanglement scores also lead to higher statistical efficiency. We note that some AnnealedVAE models seem to exhibit a high statistical efficiency on Scream-dSprites and to some degree on Noisy-dSprites. This can be explained by the fact that these models have low downstream performance and that hence the accuracy with 100 samples is similar to the accuracy with 10 000 samples. We further observe that DCI Disentanglement and MIG seem to be lead to a better statistical efficiency on the the data set Shapes3D for gradient boosted trees. Figures 22 and 23 show the downstream performance for three groups with increasing levels of disentanglement (measured in DCI Disentanglement and MIG respectively). We observe that indeed models with higher disentanglement scores seem to exhibit better performance for gradient boosted trees with 100 samples. However, considering all data sets, it appears that overall increased disentanglement is rather correlated with better downstream performance (on some data sets) and not statistical efficiency. We do not observe that higher disentanglement scores reliably lead to a higher sample efficiency.

**Implications.**   While the empirical results in this section are negative, they should also be interpreted with care. After all, we have seen in previous sections that the models considered in this study fail to reliably produce disentangled representations. Hence, the results in this section might change if one were to consider a different set of models, for example semi-supervised or fully supervised one. Furthermore, there are many more potential notions of usefulness such as interpretability and fairness that we have not considered in our experimental evaluation. Nevertheless, we argue that the lack of concrete examples of useful disentangled representations necessitates that future work on disentanglement methods should make this point more explicit. While prior work [54, 34, 41, 20, 21] successfully applied disentanglement methods such as $\beta$-VAE on a variety of downstream tasks, it is not clear to us that these approaches and trained models performed well *because of disentanglement*.

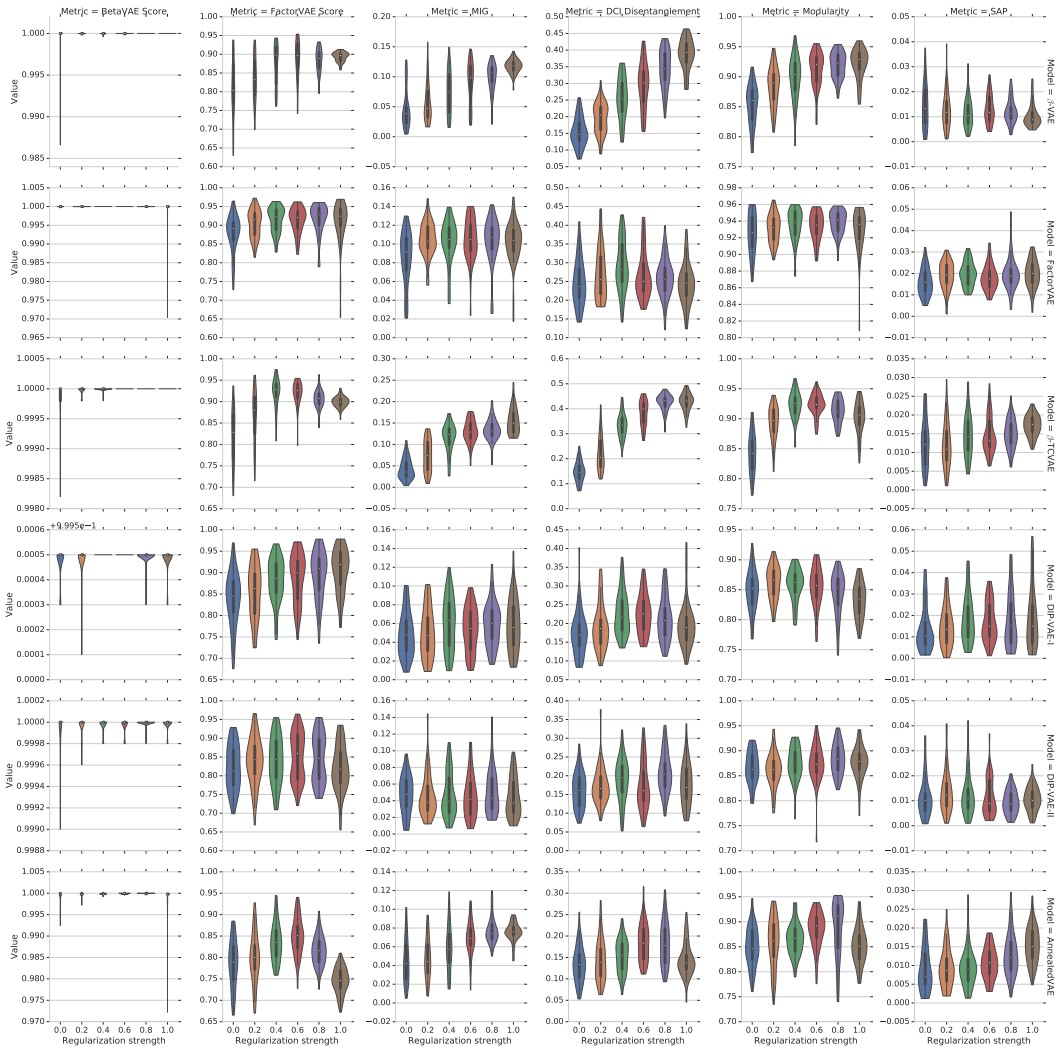

Figure 14: Distribution of scores for different models, hyperparameters and regularization strengths on Cars3D. We clearly see that randomness (in the form of different random seeds) has a substantial impact on the attained result and that a good run with a bad hyperparameter can beat a bad run with a good hyperparameter in many cases.

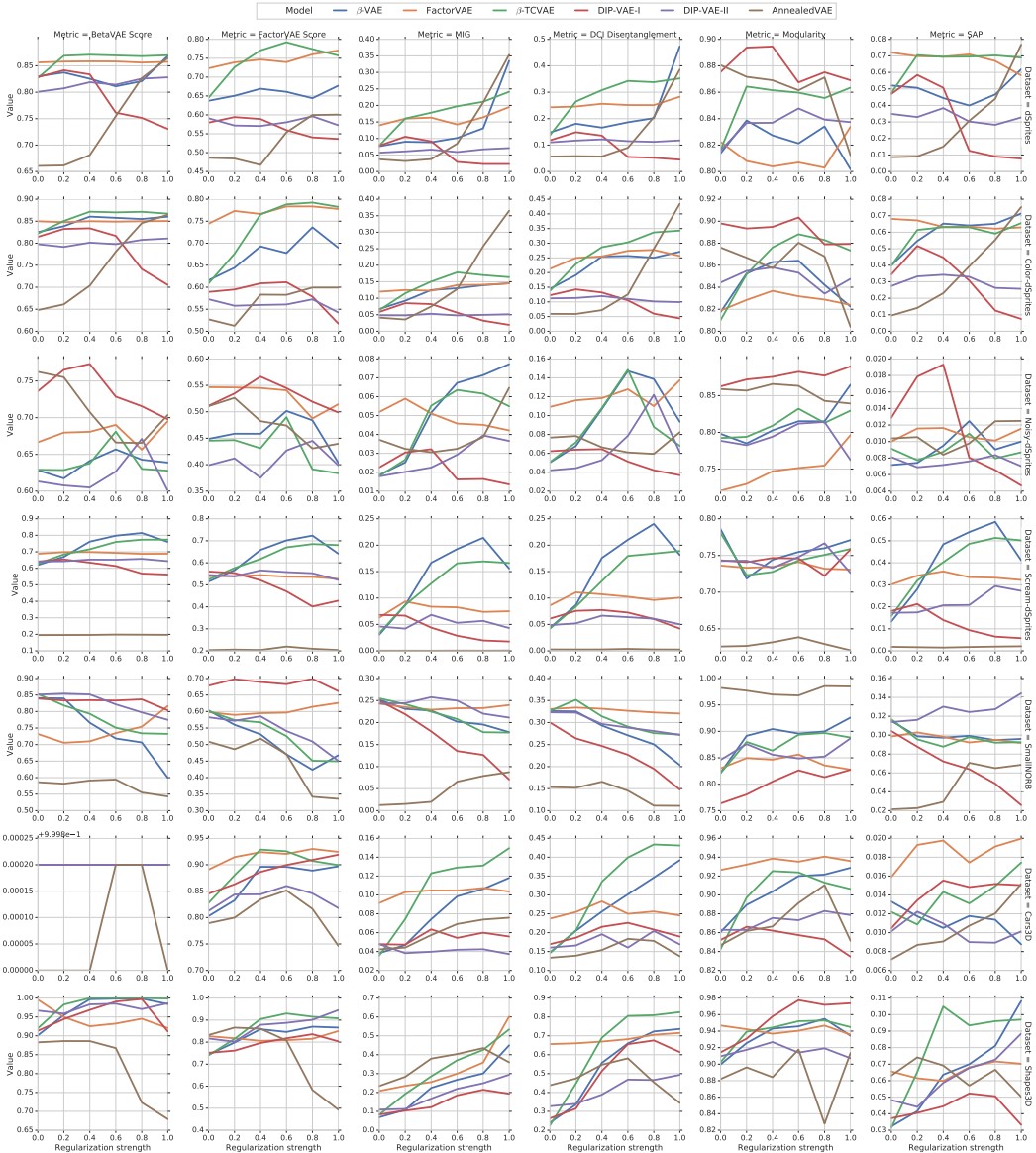

Figure 15: Score vs hyperparameters for each score (column) and data set (row). There seems to be no model dominating all the others and for each model there does not seem to be a consistent strategy in choosing the regularization strength.

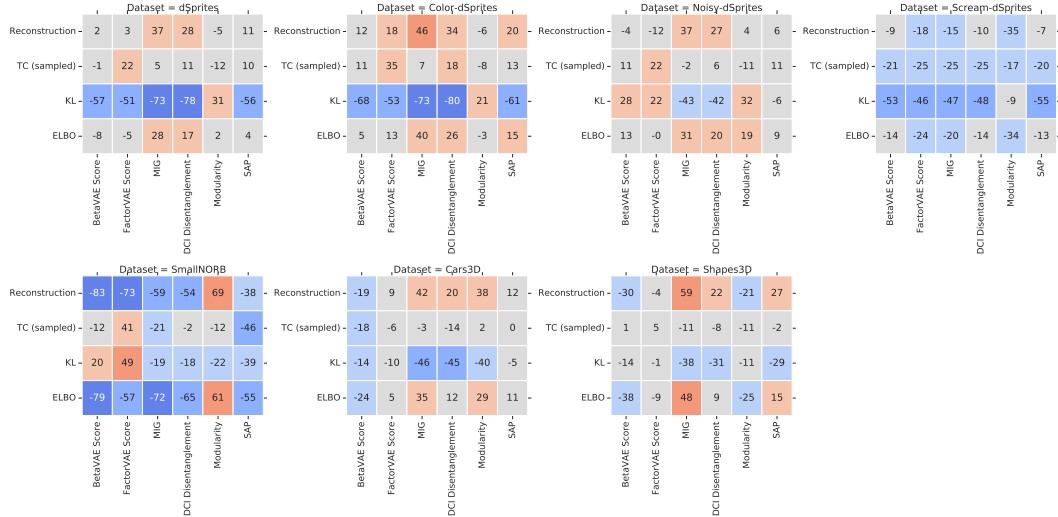

Figure 16: Rank correlation between unsupervised scores and supervised disentanglement metrics. The unsupervised scores we consider do not seem to be useful for model selection.

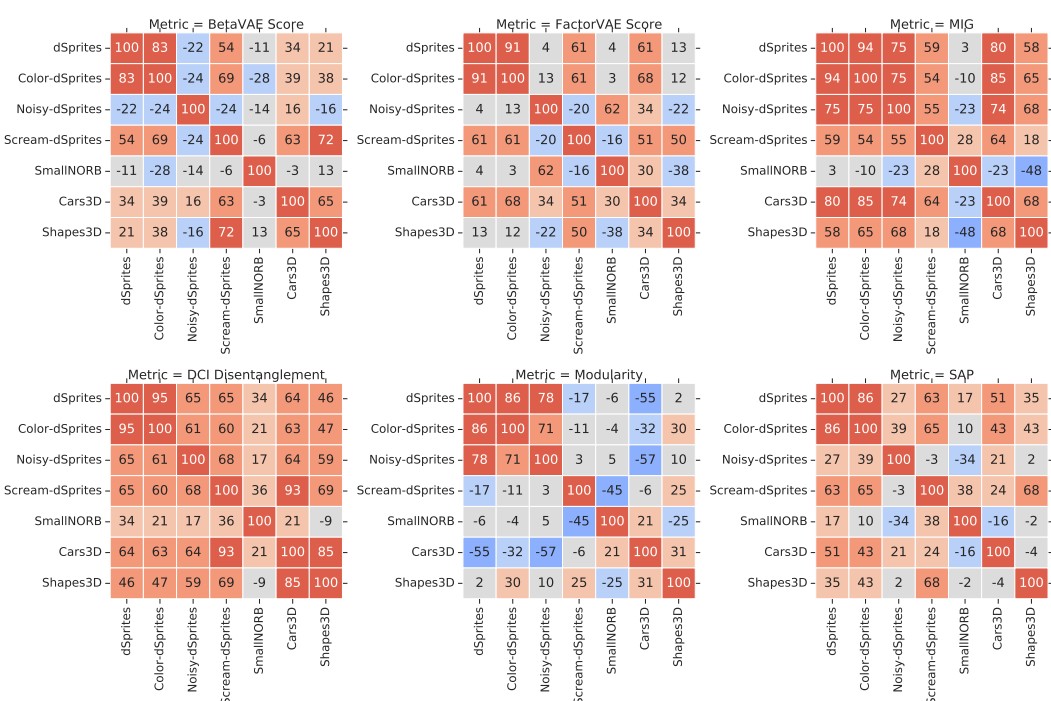

Figure 17: Rank-correlation of different disentanglement metrics across different data sets. Good hyperparameters only seem to transfer between dSprites and Color-dSprites but not in between the other data sets.

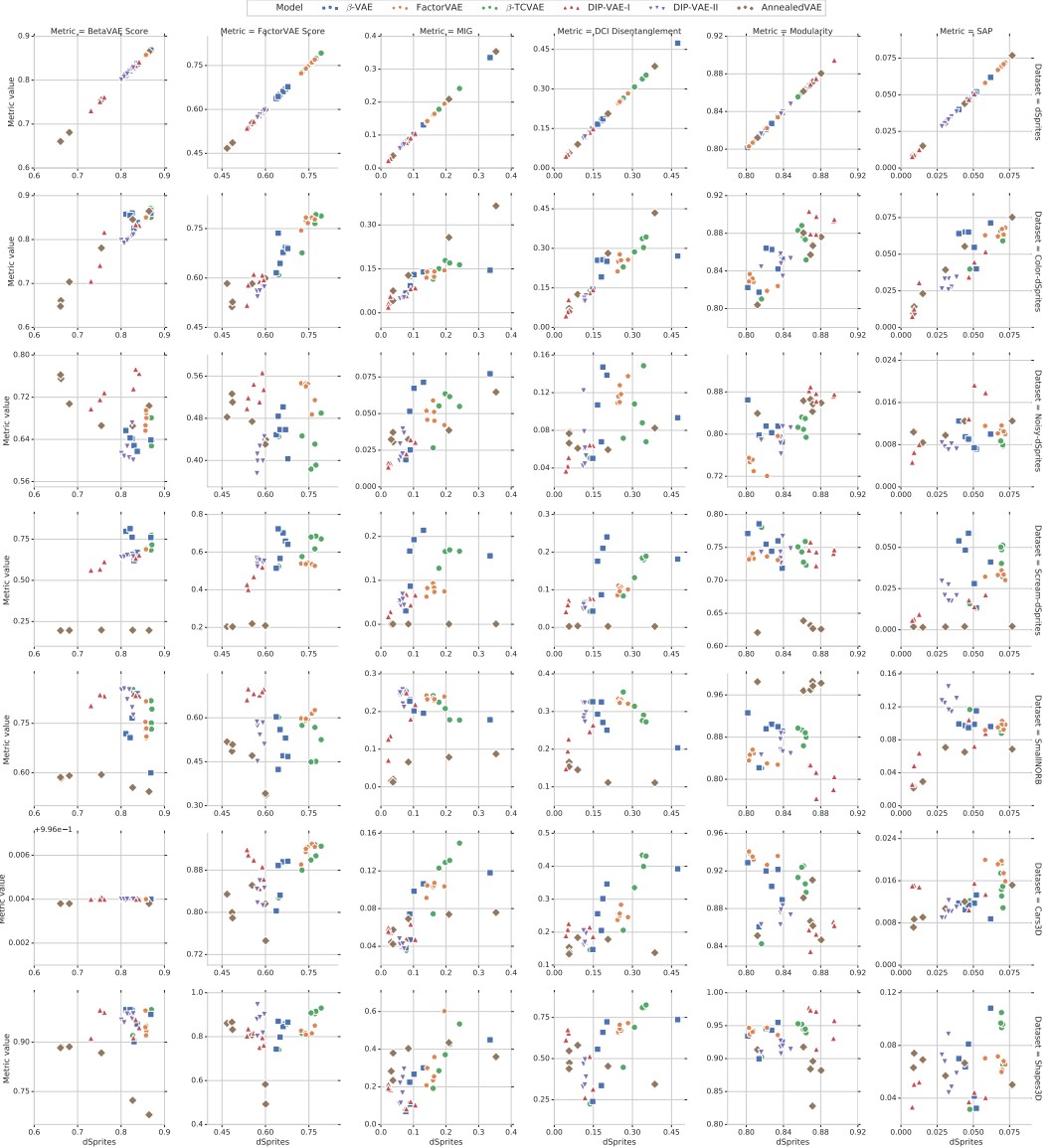

Figure 18: Disentanglement scores on dSprites vs other data sets. Good hyperparameters only seem to transfer consistently from dSprites to Color-dSprites.

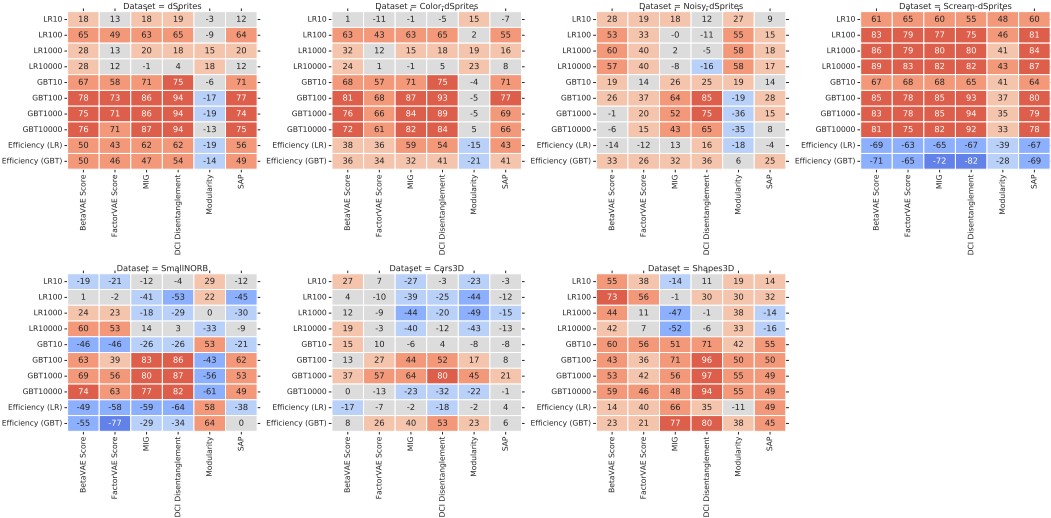

Figure 19: Rank-correlation between the metrics and the performance on downstream task on different data sets. We observe some correlation between most disentanglement metrics and downstream performance. However, the correlation varies across data sets.

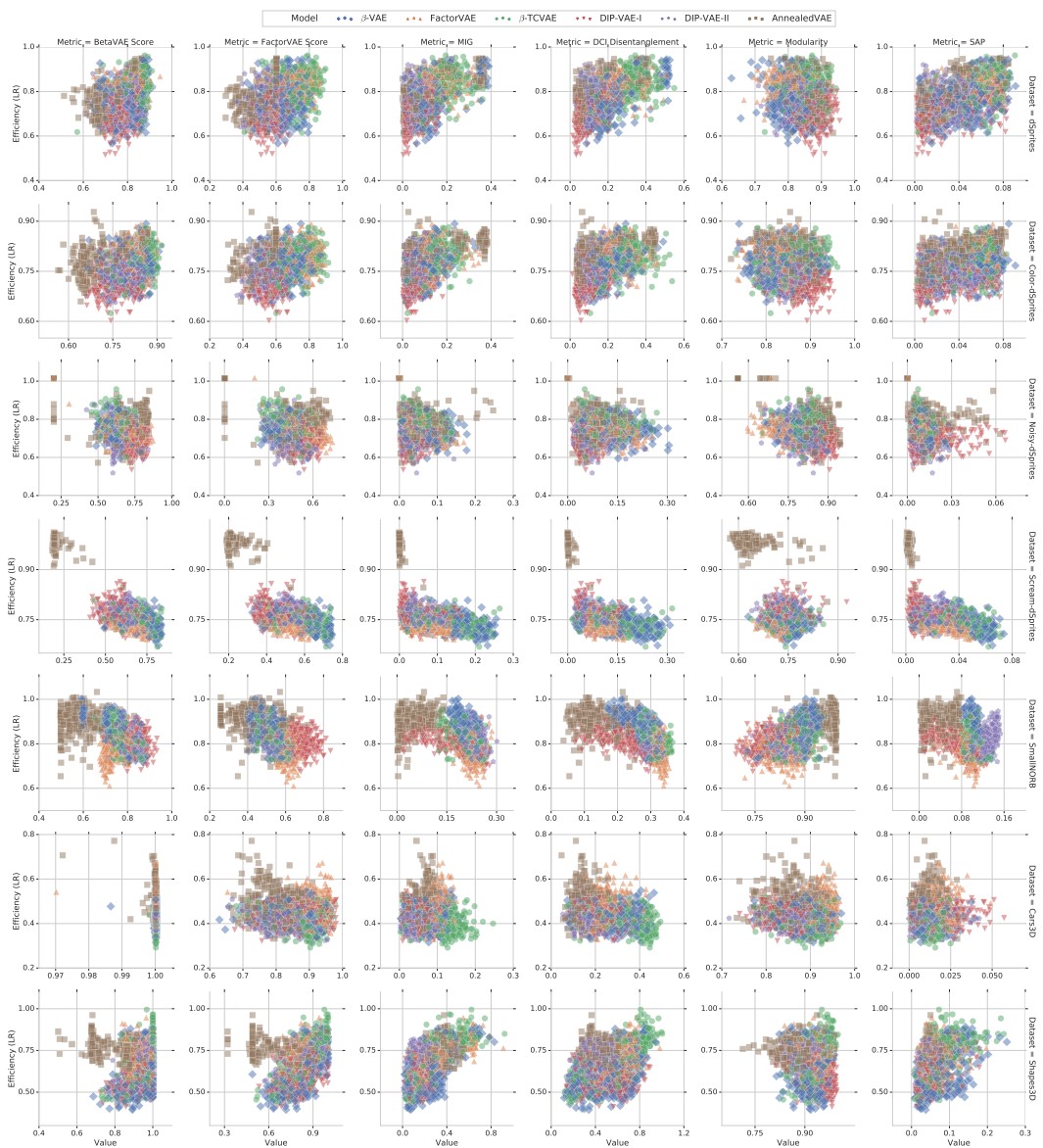

Figure 20: Statistical efficiency (accuracy with 100 samples ÷ accuracy with 10 000 samples) based on a logistic regression versus disentanglement metrics for different models and data sets. We do not observe that higher disentanglement scores lead to higher statistical efficiency.

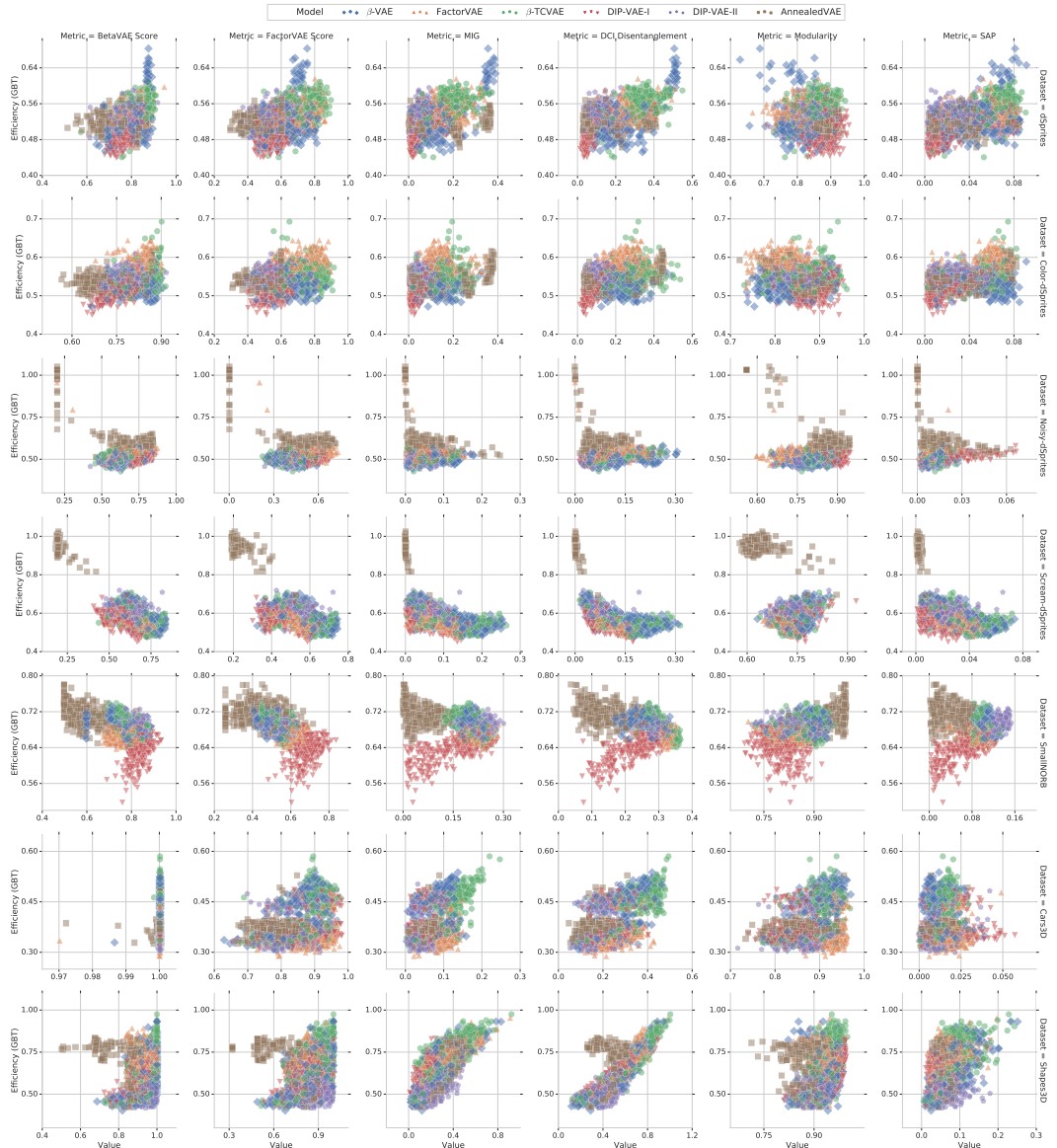

Figure 21: Statistical efficiency (accuracy with 100 samples ÷ accuracy with 10 000 samples) based on gradient boosted trees versus disentanglement metrics for different models and data sets. We do not observe that higher disentanglement scores lead to higher statistical efficiency (except for DCI Disentanglement and Mutual Information Gap on Shapes3D and to some extend in Cars3D).

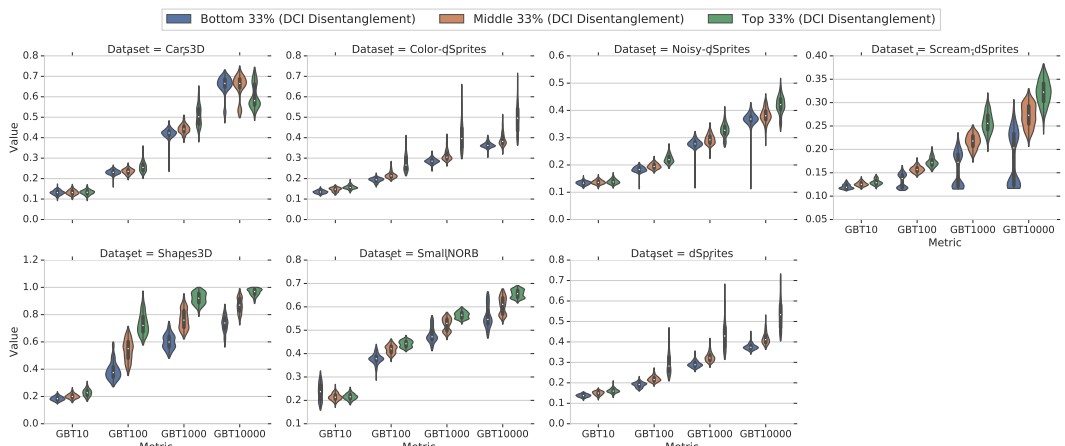

Figure 22: Downstream performance for three groups with increasing DCI Disentanglement scores.

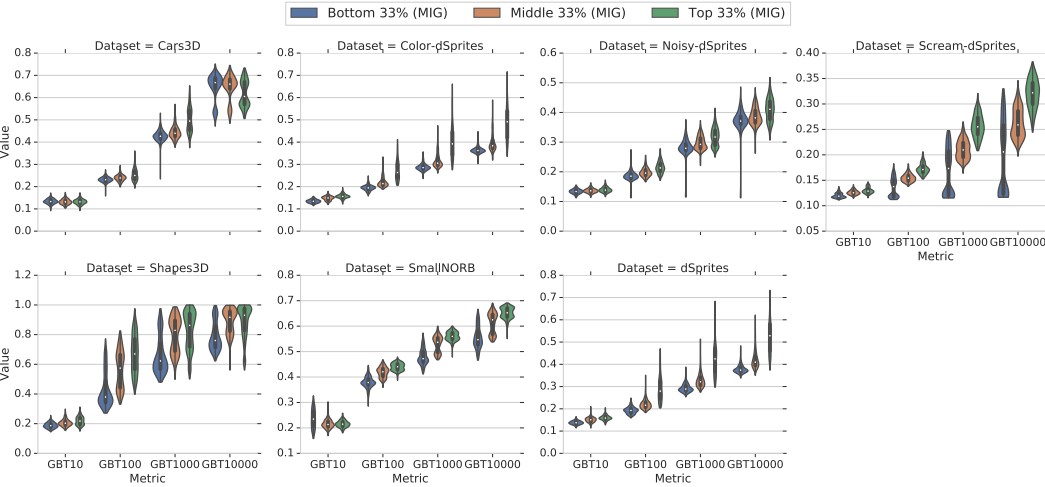

Figure 23: Downstream performance for three groups with increasing MIG scores.

# K Additional Figures

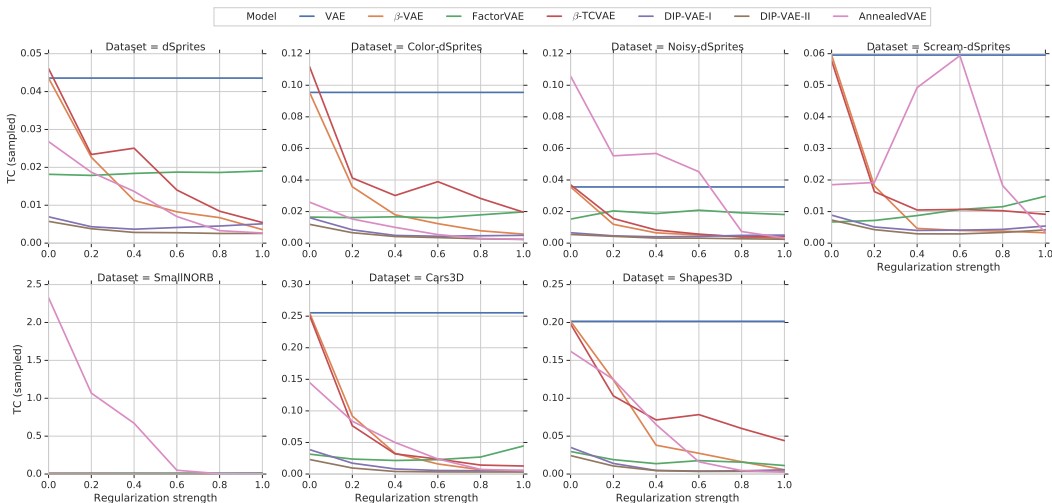

Figure 24: Total correlation of sampled representation plotted against regularization strength for different data sets and approaches (including AnnealedVAE).

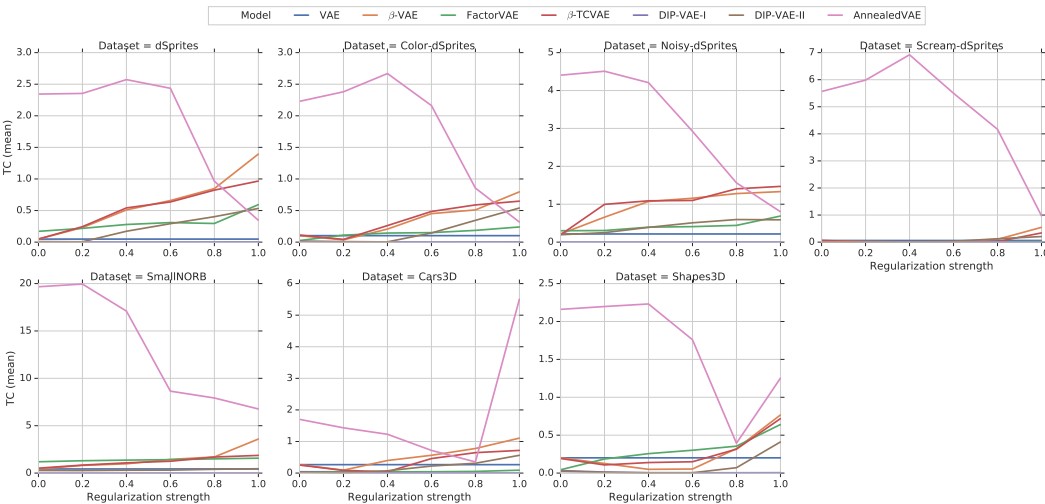

Figure 25: Total correlation of mean representation plotted against regularization strength for different data sets and approaches (including AnnealedVAE).

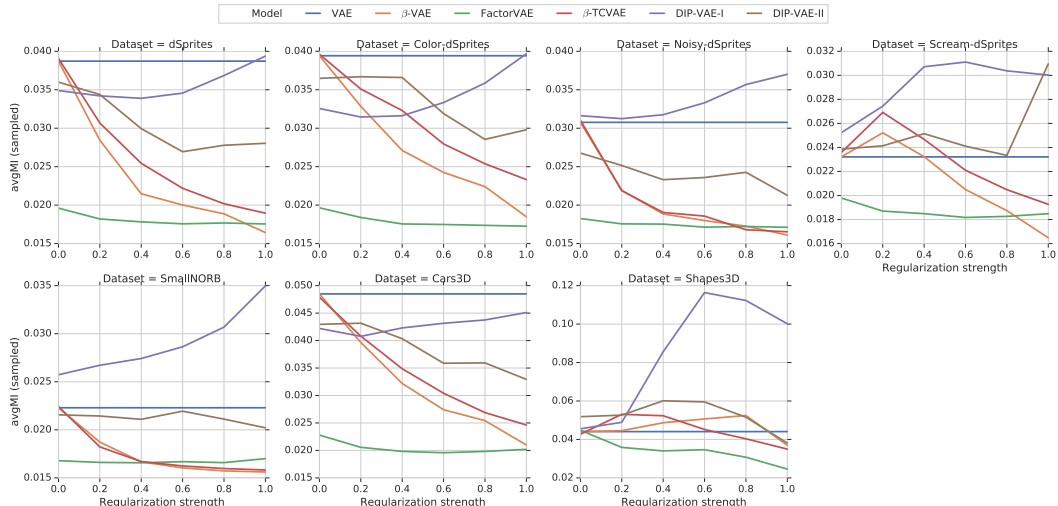

Figure 26: The average mutual information of the dimensions of the sampled representation generally decrease except for DIP-VAE-I.

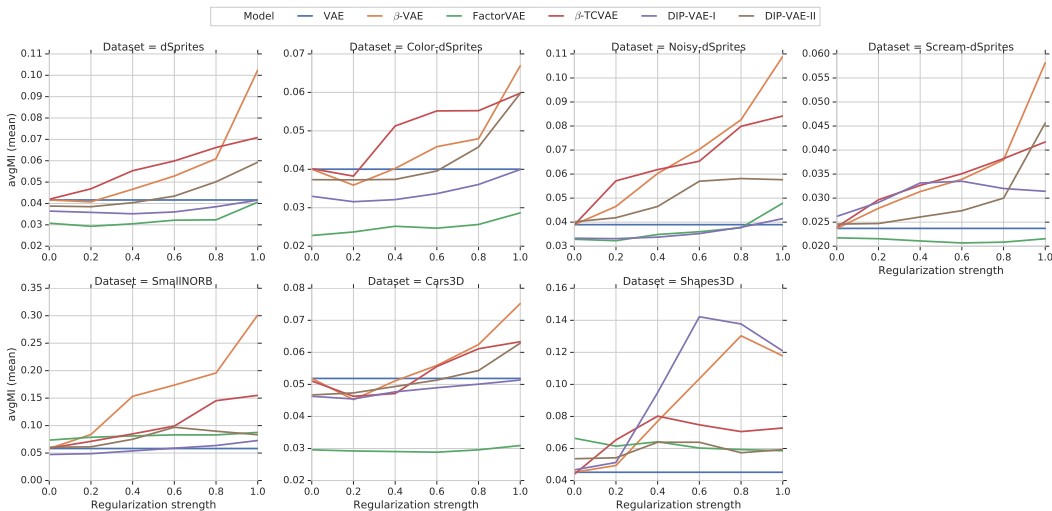

Figure 27: The average mutual information of the dimensions of the mean representation generally increase.

