# OpenReview forum: "Challenging Common Assumptions in the Unsupervised Learning of Disentangled Representations"
_ICLR.cc/2019/Workshop/RML — RML 2019_

### Official Review · AnonReviewer1 · 2019-04-01
**Interesting work on representation learning**

**Rating:** 4
**Confidence:** 3

**Review:**

Summary: This paper is an interesting discussion of disentanglement with an experimental study on several VAE variants.  I strongly recommend acceptance, however the writing could be a bit more conservative in the claims given that only VAE-based latent variable models are studied experimentally.

Notes:
  -This paper investigates many proposed methods for learning disentangled representations.
  -Introduction does a good job of laying out the intuition for what we want to get out of “disentangled features”.
  -There is some intuition that changing some “disentangled features” should only change those factors of variation and not others.
  -Appendix A proves that unsupervised learning of disentangled representations is impossible without inductive biases (this doesn’t seem obvious to me!)
  -Paper measures disentanglement across 12k models.
  -Releases a “disentanglement_lib” for evaluating disentangled representations.
  -Study of different models shows that the “aggregated posterior” is not correlated, but the dimensions of the representation are correlated.  In this sentence I’m a bit confused about whether this is referring to q(z|x) or q(z).  On first reading, I find this claim a bit confusing, because if q(z) follows a gaussian distribution, then its dimensions should be disentangled?
  -All datasets considered work on the assumption that x is a deterministic function of an underlying disentangled z.

Comments:
  -Uses the wrong style sheet.
  -It’s a bit weird to play up the “10k models” aspect, because presumably this comes from some kind of hyperparameter sweep or combinatorial explosion?
  -I think it seems like kind of a bad omission to not include ALI (Dumoulin 2016) or any other models in this family.
  -Beginning of 4.3 has a typo.

---

### Decision · Program_Chairs · 2019-04-05
**Acceptance Decision**

Accept